# Tracing metal footprints via global renewable power value chains

Rao Fu[1,8], Kun Peng[1,8], Peng Wang[2], Honglin Zhong[1,3], Bin Chen[4], Pengfei Zhang[1], Yiyi Zhang [5], Dongyang Chen[6], Xi Liu[1] ✉, Kuishuang Feng [7] ✉ & Jiashuo Li [1,3] ✉

The globally booming renewable power industry has stimulated an unprecedented interest in metals as key infrastructure components. Many economies with different endowments and levels of technology participate in various production stages and cultivate value in global renewable power industry production networks, known as global renewable power value chains (RPVCs), complicating the identification of metal supply for the subsequent low-carbon power generation and demand. Here, we use a multi-regional input-output model (MRIO) combined with a value chain decomposition model to trace the metal footprints (MFs) and value-added of major global economies' renewable power sectors. We find that the MFs of the global renewable power demand increased by 97% during 2005−2015. Developed economies occupy the high-end segments of RPVCs while allocating metal-intensive (but low value-added) production activities to developing economies. The fast-growing demand for renewable power in developed economies or developing economies with upper middle income, particularly China, is a major contributor to the embodied metal transfer increment within RPVCs, which is partly offset by the declining metal intensities in developing economies. Therefore, it is urgent to establish a metal-efficient and green supply chain for upstream suppliers as well as downstream renewable power installers for just transition in the power sector across the globe.

Renewable power is one of the best options for a more sustainable energy system that would allow our society to reduce man-made greenhouse gas emissions or meet intended climate goals, such as the Paris Agreement[1–3]. However, the infrastructure (solar modules, wind turbines, etc.) of renewable power relies on various metals such as iron, copper, aluminum, and other precious metals[4–6]. In the current globalized world, economies play different roles in renewable power supply chains ranging from mining, refinery, and component manufacturing, on to the final deployment, and generate revenues (value-added) in each stage of the renewable power value chains (RPVCs)[7–10]. For instance, solar photovoltaic (PV) value chains use metal ores (copper, aluminum, etc.) from China[11] and Africa[12,13], and modules (silver, copper, etc.) from Europe[14], the United States[15], and China[16,17], which are then assembled in developing economies in Asia, and sold globally[18]. With the rapid expansion of renewable power and the increasing complexity of RPVCs, it is increasingly challenging to

[1]Institute of Blue and Green Development, Shandong University, Weihai 264209, P. R. China. [2]Key Lab of Urban Environment and Health, Institute of Urban Environment, Chinese Academy of Sciences, Xiamen 361021, P. R. China. [3]Academy of Plateau Science and Sustainability, Qinghai Normal University, Xining 810016, P. R. China. [4]Department of Environmental Science and Engineering, Fudan University, Shanghai 200082, P. R. China. [5]Guangxi Key Laboratory of Power System Optimization and Energy Technology, Guangxi University, Nanning 530004, P. R. China. [6]Shandong Key Laboratory of Blockchain Finance, Shandong University of Finance and Economics, Jinan 250014, P. R. China. [7]Department of Geographical Sciences, University of Maryland, College Park, MD 20742, USA. [8]These authors contributed equally: Rao Fu, Kun Peng. ✉e-mail: liuxi@sdu.edu.cn; kfeng@umd.edu; lijiashuo@sdu.edu.cn

identify the metal product suppliers of the renewable power sector[19,20]. Clearly tracing how RPVCs affect metal use or value-added can provide valuable information for policymakers to formulate trade policies and foster sustainable and responsible RPVCs.

The key to solving the difficulties in understanding who supplies metal equipment for renewable power generation and demand lies in identification of the links between RPVCs and metal demand. Currently, some research focused on estimating the metal demand and constraints of the renewable power sectors. For example, Wang et al. found that the cumulative amount of critical metals required for the production of China's solar power from 2015 to 2050 will exceed the present national reserve by 1.4–123 folds[11]. Similar studies have been conducted regarding wind power[10,21], hydroelectricity[22,23], and nuclear power-related[24,25] metal demands. Most previous estimates focused on direct metal use[10,21,26–31], while a more comprehensive assessment regarding upstream metal applications related to supply chain activities (such as transportation and service) is scarce[32–34].

In the context of an internationally fragmented renewable power production network, more economies and industries are involved and connected. Although previous studies have estimated the region-specific metal demands for renewable power sectors, they lack a detailed and comprehensive picture of the interactions between RPVCs and metal use[35]. This is because these previous studies generally neglect metal use associated with critical upstream stages or do not identify how production sharing affects metal use or value-added among economies in global RPVCs. Therefore, there is an urgent need to conduct a detailed evaluations of both direct and indirect metal use or value-added of different production stages along RPVCs, given that crucial information such as the metal costs induced by renewable power and the position of each economy along RPVCs for reasonably allocating metal use responsibility remains poorly understood[36–39]. Moreover, the renewable power sectors' scale and the trade of renewable power products have witnessed substantial growth over the last two decades (e.g., the solar photovoltaic module imports of the United States increased by ~70 times in the past 15 years)[40,41], which inevitably changes the profiles of metals consumption as well as metals embodied in international trade[42]. To this end, it is vital to unveil the evolution trajectory of metal demand induced by renewable power and the driving forces behind it, which is essential for promoting cross-boundary joint actions for efficient metal use in RPVCs.

In this work, we develop a quantitative framework to gauge metal footprints (MFs, the total metal ores embodied in RPVCs) in global RPVCs by combining a multi-regional input-output model (MRIO) with a value chain decomposition model. The model enables us to track the direct and indirect metal use or value-added associated with all supply chain activities. Specifically, we focus on widely recognized crucial metal demand[31,43,44] for seven categories of renewable power sector (hydropower, wind power, bioenergy, solar PV, solar thermal, ocean power and geothermal power). The metals are grouped into four categories, as suggested in the reports of the United Nations Environment Programme (UNEP) and Word Bank[45,46]: bulk metal ores (bauxite, copper, iron, lead, and zinc ores), precious metal ores (silver and platinum group metal ores), scarce metal ores (nickel and tin ores), and other non-ferrous metal ores. We use a highly detailed MRIO table to trace the spatial-temporal changes in renewable power sectors' MFs and value-added in 49 economies during 2005–2015. Furthermore, the value chain status of each economy is presented by comparing domestic metal use embodied in exports with the corresponding domestic value-added (see Methods). Consequently, we provide a more holistic view of the growing imbalances in economic benefits and metal costs within RPVCs, highlighting the urgent need to formulate appropriate responsible strategies. In addition, the structural decomposition analysis (SDA) model is applied to investigate the differentiated contribution of each domestic and foreign driving factor to embodied metal changes in trade. By doing so, we reveal the driving

mechanism of growing MFs inequality, enabling decision makers and practitioners to formulate targeted measures and policies for mitigating potential growing metal inequities and efficient metal use in RPVCs.

## Results
### Global and regional metal footprints of renewable power demand
The evolutionary trends in the MFs of the global renewable power sector are shown in Fig. 1 and Supplementary Fig. 1a, b. Along with the rapid expansion of renewable power infrastructure worldwide, the MFs of the global renewable power demand increased by 97% or 2425 kilotons (kt) from 2005 to 2015. Comparatively, the installed capacity of renewable power is accelerating, increased by 125% or 1101 gigawatts (GW) over 10 years. Both MFs and installed capacity in the wind power and solar PV sectors experienced a large increasing rate (3.2–3.5 times of the MFs and 6–46 times of the installed capacity for renewable power). All results indicate that the installed capacity and electricity generation are growing faster than the MFs of the renewable power demand, mainly because of the material efficiency improvement of renewable power technology in its whole supply chain. There are various options for improving material efficiency, such as design optimization[47], size enlargement[31], and material substitution[48].

Significant discrepancies exist in the MFs of different economies because of the vast differences in the scale of their renewable power sectors and their metal use efficiency. China was the global leader in renewable power sector scale with 67.7 GW newly added installed capacity (43% of the global total) and 1381 terawatt hours (TWh) of renewable electricity generation (24% of the global total), accounting for 61% (Supplementary Fig. 1c) of the total MFs of the global renewable power demand in 2015. In comparison, the United States, the second-largest global economy in renewable power sector scale, added 17.3 GW of renewable power capacity (11% of the global total) and generated 568 TWh of renewable electricity (9% of the global total), accounting for only 1% (Supplementary Fig. 1c) of the global renewable power sectors' MFs. One complicating factor is that the intensity of metal use (kilogram/megawatt hours, kg/MWh, the ratio of total supply chain metal use to gross renewable electricity generation of each economy, Supplementary Table 1) in the renewable sector of the United States (0.09 kg/MWh) is much lower than that of China (2.2 kg/MWh). In addition, developing economies, such as Latin America, Other Asia, and Africa held 11% of the global MFs of renewable power demand, with only 6% of the global renewable power installations in 2015. This is mainly due to the high intensity of metal use for renewable power generation in these developing economies.

In general, iron and copper ores were the dominant metals in the production of global renewable power, accounting for 54 and 26% of the total MFs of the global renewable power demand, respectively (Supplementary Fig. 2), followed by other non-ferrous metal ores (6% of global MFs). Figure 1 and Supplementary Fig. 3 further show that metal composition of the same renewable power sector is considerably different across economies. Iron ores constituted more than half of the MFs of wind power demand in China, whereas copper appeared to be the dominant metal in Poland and the United Kingdom. The difference can be attributed to differentiated upstream sectoral structures among the economies (as shown in Supplementary Fig. 4). Indirect metal use contributed to more than 80% of the total metal ores use for per unit of wind power generation in the three economies. In China, iron ores accounted for nearly two-thirds of the indirect metal use, mainly induced by the electrical machinery, equipment, and fabricated metal products manufacturing sectors, which are iron ore-intensive processes from a supply chain perspective. The United Kingdom and Poland, exhibited a large proportion of copper ores occupancy, mainly induced by the copper ore-intensive processes through the entire supply chain, such as construction or copper

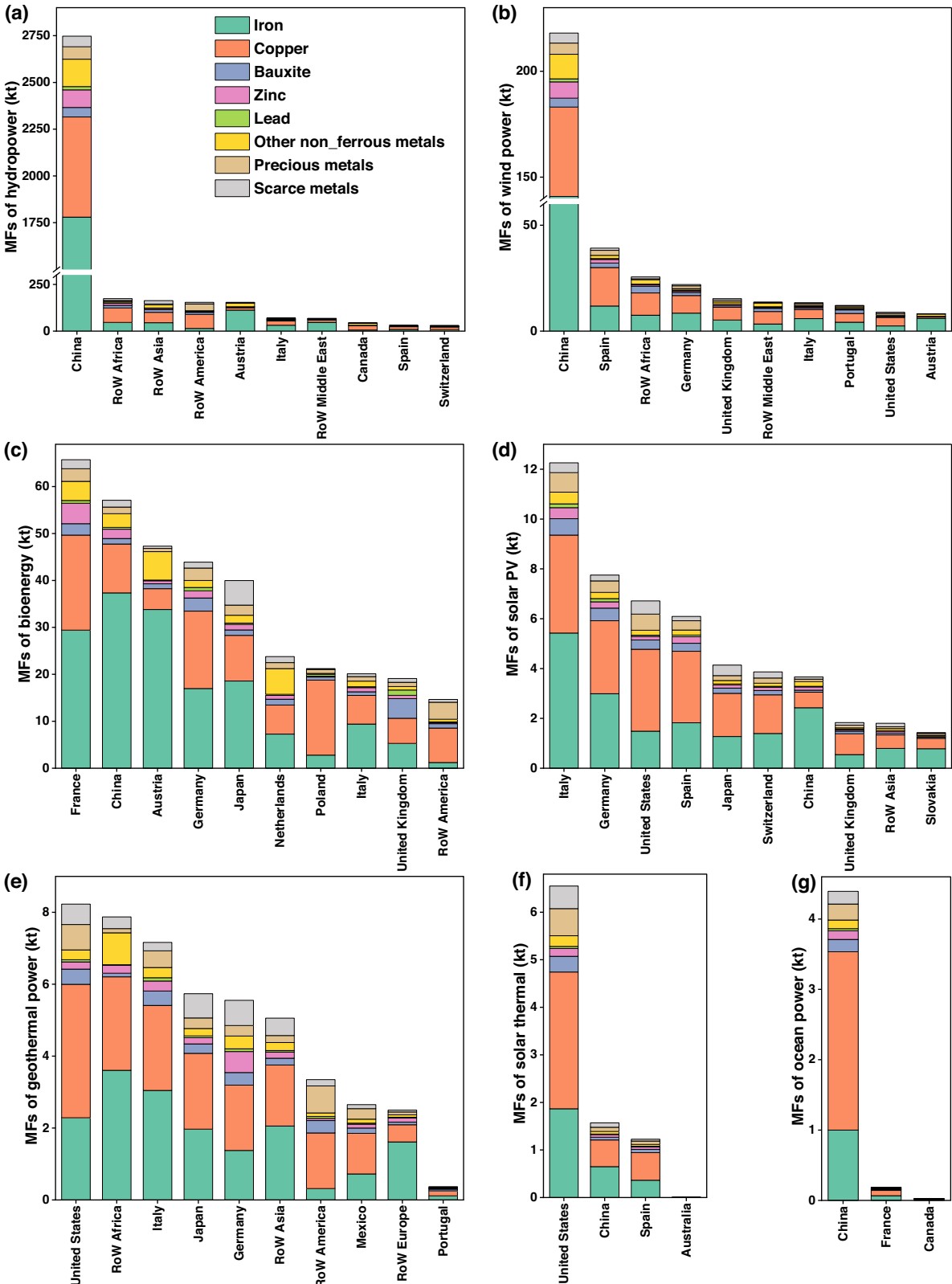

**Fig. 1 | Metal footprints (MFs) of the renewable power demand by metal types in 2015.** Top ten economies in MFs of **a** hydropower, **b** wind power, **c** bioenergy, **d** solar photovoltaic (PV), **e** geothermal power, **f** solar thermal, **g** ocean power.

Among all economies, the top 10 economies with more than half the global MFs for each renewable power demand are included unless the MFs = 0 (for additional economy MFs, see Supplementary Fig. 3).

mining activities. Likewise, when comparing China with RoW America (Latin American economies, excluding Brazil), hydropower presents distinct patterns, detailed explanations are presented in Supplementary Fig. 4 and Supplementary Section B.1.

**Outsourced metal footprints in RPVCs**

Metal outsourcing indicates that an economy increases metal ores extraction outside its borders for domestic consumption of renewable power. From the period of 2005 to 2015, there was a notable increase

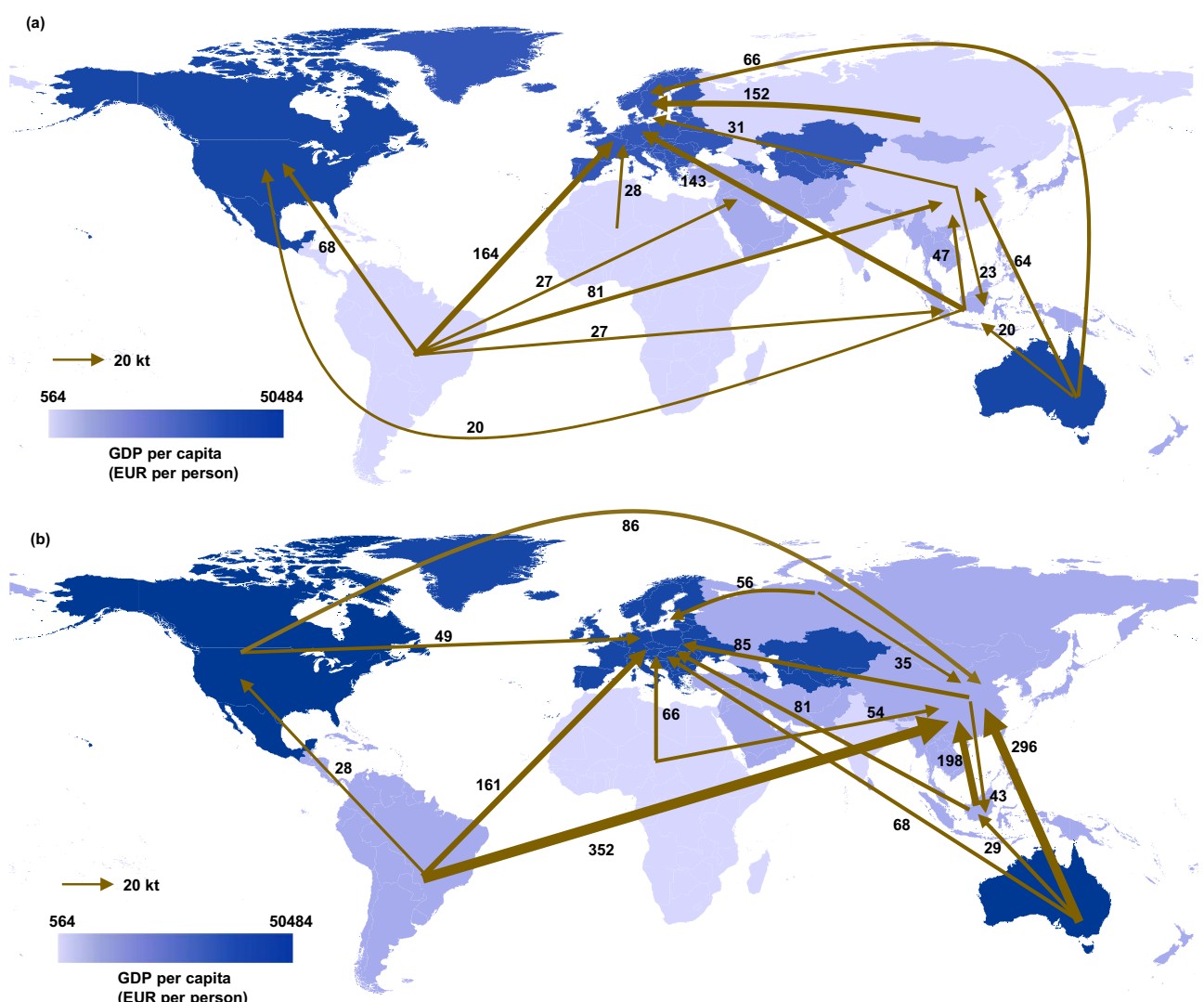

**Fig. 2 | Major international flows of metals (>20 kt) embodied in renewable power value chains (RPVCs) among ten groups of economies. a** 2005, **b** 2015, the economies are shaded according to value of Gross Domestic Product (GDP) per capita. The arrows indicate the direction and magnitude of foreign metal ores embodied in renewable power consumed by destination economy. Nearly 80% of global total flows are shown, with the rest flows presented in Supplementary Tables 3 and 4.

in the outsourcing of metals, an escalation of 69%, equivalent to 833 kilotons. This surge was predominantly instigated by economies that are either developed or are classified as upper middle-income developing economies. China, in particular, stands out as a major contributor in this context (refer to Fig. 2 for details). The categorization of developed and developing economies is drawn from the classification provided by the World Bank[49] (Supplementary Table 2). In RPVCs, developed economies outsource increasingly large amounts of metal consumption for renewable power sectors to developing economies, which leads to an increase in metal mining and production in developing economies. Here, we aggregated the results into 10 groups of economies (Europe, Africa, the Middle East, North America, Latin America, Other Asia, China, India, Russia, and Australia) to clarify the outsourced MFs flow patterns.

Europe represented the major importer of metals, with 29–52% of the total global import volume during the period concerned (Fig. 2). In 2015, Europe imported 598 kt metals, mostly from Latin America (160 kt), Other Asia (82 kt), and Africa (66 kt). This is mainly attributable to the shortage of indigenous mineral resources, such as copper and lead ores, which cannot meet the large metal demand induced by the huge renewable power industry[50]. As a result, European economies must turn to large metal and renewable power

component exporters for imports. Interestingly, China quickly became the largest importer (264 kt in 2005 and 1108 kt in 2015), with imported metals mainly from Latin America (352 kt), Australia (296 kt), and Other Asia (198 kt) in 2015. Unlike the developed economy, >63% of the metals consumed by China's renewable power demand was satisfied by domestic supply (Supplementary Fig. 5). Although China is rich in several metal endowments (such as zinc), it remains insufficient to meet the fast-growing renewable power industry's metal demand, motivated by carbon mitigation ambition[51]. In particular, some crucial metals for renewable power technology are inadequate, for example, scarce nickel mineral resources with <5% of global production. As a result, ~85% of the nickel footprint of the renewable power demand in China was outsourced from other economies, such as Other Asia. Comparatively, the United States increasingly relied on acquisition abroad, with the ratios of embodied metals imports to their MFs increasing from 90% in 2005 to 98% in 2015, although it contained rich indigenous mineral resources. This is because the United States shifted its manufacturing base overseas to seek greater economic and environmental benefits from trade and the integration of supply chains[18]. The United States imported 62 kt and 21 kt metals from Latin America in 2005 and 2015, respectively.

From the export perspective, more than half of the metal ores embodied in trade originated from developing economies, such as Latin America, Africa, and Other Asia (Fig. 2). Embodied metals exports from these regions accounted for 52–55% of global exports in the renewable power sector from 2005 to 2015. Among the three economies, Latin America was the largest exporter of primary products, exporting substantial amounts of metals, accounting for 29–31% of the total traded metals during 2005–2015. Moreover, Russia and Australia (together 26% of the total transfer in 2015) were also significant metal exporters that support global renewable power infrastructure.

## The positions of economies in the RPVCs

Each global economy participates in RPVCs at different production stages and generates distinct economic gains with varying metal costs. From a global value chain perspective, exports of goods to other economies entails the utilization of both domestic production and imports, resulting in the metal use and value-added generation in both domestic and foreign economies. In this research, we undertake a further decomposition of the total metal use or value-added embodied in exports, aimed at fulfilling the demand for renewable power in foreign economies. This decomposition separates domestic and foreign components to highlight their distinctive positions. To this end, we introduce two indicators, namely, the domestic metal use ratio (DMUR) and domestic value-added ratio (DVAR) of total exports for satisfying foreign renewable power demand (depicted in Fig. 3a, b and Supplementary Fig. 6). A higher DMUR or DVAR indicates that the majority of metal use or value-added triggered by foreign renewable power demand occurs domestically. In contrast, an economy with a large DMUR but a small DVAR implies that the economy has a considerable contribution of metal use to fulfill foreign renewable power demand but gains a minimal economic benefit.

In general, developed economies occupied high-end segments of the RPVCs. Developed economies, such as European economies (Fig. 3a, b), which exported high-tech and high-value-added intermediate products, consumed the least domestic metals (Fig. 3c). This is because developed economies tended to have high-tech sectors and add a large amount of value through high-end manufacturing or design stages, which consume low levels of metals. As shown in Fig. 3a, b, Norway, Germany, and the Netherlands contributed 42% of global value-added, with a large share of their domestic value-added (DVAR more than 85% in 2015) while consuming far below average (the horizontal dotted line) of the metals extracted locally. With low domestic metals consumption–high-value-added, these economies occupied the top location in the RPVCs. In comparison, developing economies tended to export low-end, low-value-added products, such as ores and steel plates. The developing economies (e.g., Latin America and Other Asia) contributed a large proportion (DMUR ~70% in 2015, Fig. 3b) of the metals mined domestically (Fig. 3c) but received the least value-added (2–7% of the world's total) for satisfying foreign renewable power demand. Because they have the lowest production costs and the least strict environmental regulations in the world, developing economies have become the destination for manufacturing processes outsourced from developed economies[46]. Interestingly, China, as the world's top renewable power installer, held 2.6% of the global total value-added (Fig. 3) induced by goods and services exports to satisfy foreign renewable power demand, similar to France (2.1%), both of which were lower than that of Norway (23%). This is mainly due to the limited scale of goods and services exports in China and their low-end position in production stages, that is, exporting more domestic metals (more than 168 kt in China, Fig. 3c) with less economic gains (>595 million EUR, M.EUR). In contrast, China occupied the largest value-added created by goods and services to meet both domestic and foreign renewable power demand (Supplementary Fig. 8).

During the period from 2005 to 2015, developed economies, such as Norway, Netherlands, and Germany, sustained their prominence in global RPVCs and consistently exhibited high DVAR. These economies

gradually shifted towards higher value-added, yet less metal-intensive production stages. For example, Germany experienced a 1.1% increase in DVAR, resulting in a value-added augmentation of 58 million EUR, which constituted 0.5% of the overall global increment. Concurrently, the metal content in Germany's exports declined by 0.22 kt during the same decade. In contrast, the developing economies witnessed moderate growth in their participation in global RPVCs, characterized by slight increases in both DVAR and DMUR. Nevertheless, developing economies accounted for a substantially larger proportion of the global increase in domestic metals exports compared to the value-added gains they experienced. For instance, Latin America observed a 154.6 kt increase in metal content within its exports, representing 16.5% of the global total increase. However, the region's value-added growth was relatively modest at 1.8% of the global total increment. Similarly, China demonstrated growth in DMUR and DVAR by 2.7% and 4.9%, respectively. This growth corresponded to a 94 kt increase in metals embodied in intermediate goods exports, which comprised 10% of the global total increase. Despite this growth, the value-added increase in China was disproportionately smaller, accounting for merely 4% of the global total increase between 2005 and 2015.

## The growing metal footprints inequality and its driving forces

MFs inequality rose with developed economies' continuous outsourcing of metal demand for the renewable power sector to developing economies, which can be observed in three aspects. First, the MFs of renewable power per capita in developed European economies were generally higher than those in developing economies. For instance, the MFs in Sweden were ~14 times those in African economies in 2005 and grew to 18 times in 2015. Second, the gap between metals embodied in exports (MEE) and imports (MEI) along the global RPVCs continued to expand for economies. The net exports (MEE minus MEI) in developing economies increased by 21% during 2005–2015. Third, inequality grew from the perspective of domestic metals consumption per unit of export-induced value-added along RPVCs. The value of developing economies (e.g., Africa) was two times of developed economies (e.g., Europe) in 2005, which increased by a factor of 5 times in 2015.

We examine the driving forces of the metals embodied in trade to uncover the drivers of growing MFs inequality along global RPVCs (Fig. 4). The renewable power demand was the major force of inequality growth. Motivated by the renewable power ambition, the domestic renewable power demand ($\mathbf{y}^{(r)}$) boosted MEI growth by 39–93% in developed economies such as the United States and European economies in 2005–2015. Meanwhile, majority of the demand ($\mathbf{y}^{(-r)}$) from developed economies, e.g., European economies, and China induced substantial growth (98–196%) of MEE for developing economies, such as those in Latin America, Africa, and Other Asia. Comparatively, the changes in production technology ($\mathbf{H}^{(r)}$, $\mathbf{H}^{(-r)}$) and trade structure ($\mathbf{T}^{(r)}$) contributed to moderate growth in metal inequality (Supplementary Figs. 9 and 10 and Supplementary Section B.2). Production technology shifts caused the MEI increase in developed economies (except the United States and Austria) by a wide range of 13–200%, and MEE growth in developing economies by 4–99%.

In contrast, the declining direct sectoral metal intensity ($\mathbf{m}^{(r)}$, $\mathbf{m}^{(-r)}$) was a major factor that dampened metal inequality. The intensity declines in developing economies offset the MEI growth of developed economies by 70–75% and MEE growth of developing economies by 42–84% under rapid technological progress, most of which was higher than the global average of 52%. The direct sectoral metal intensity reduction mainly occurred in the upstream metal mining and production sectors, such as the mining of iron, copper, and precious metal ores, with a decrease rate of 27–100% in Latin America, Africa, and Other Asia from 2005 to 2015. Notably, the metal inequality growth driven by vigorous demands and other drivers could not be offset by the reduction in efficiency gains, indicating the growing imbalance among economies to support the global renewable power market.

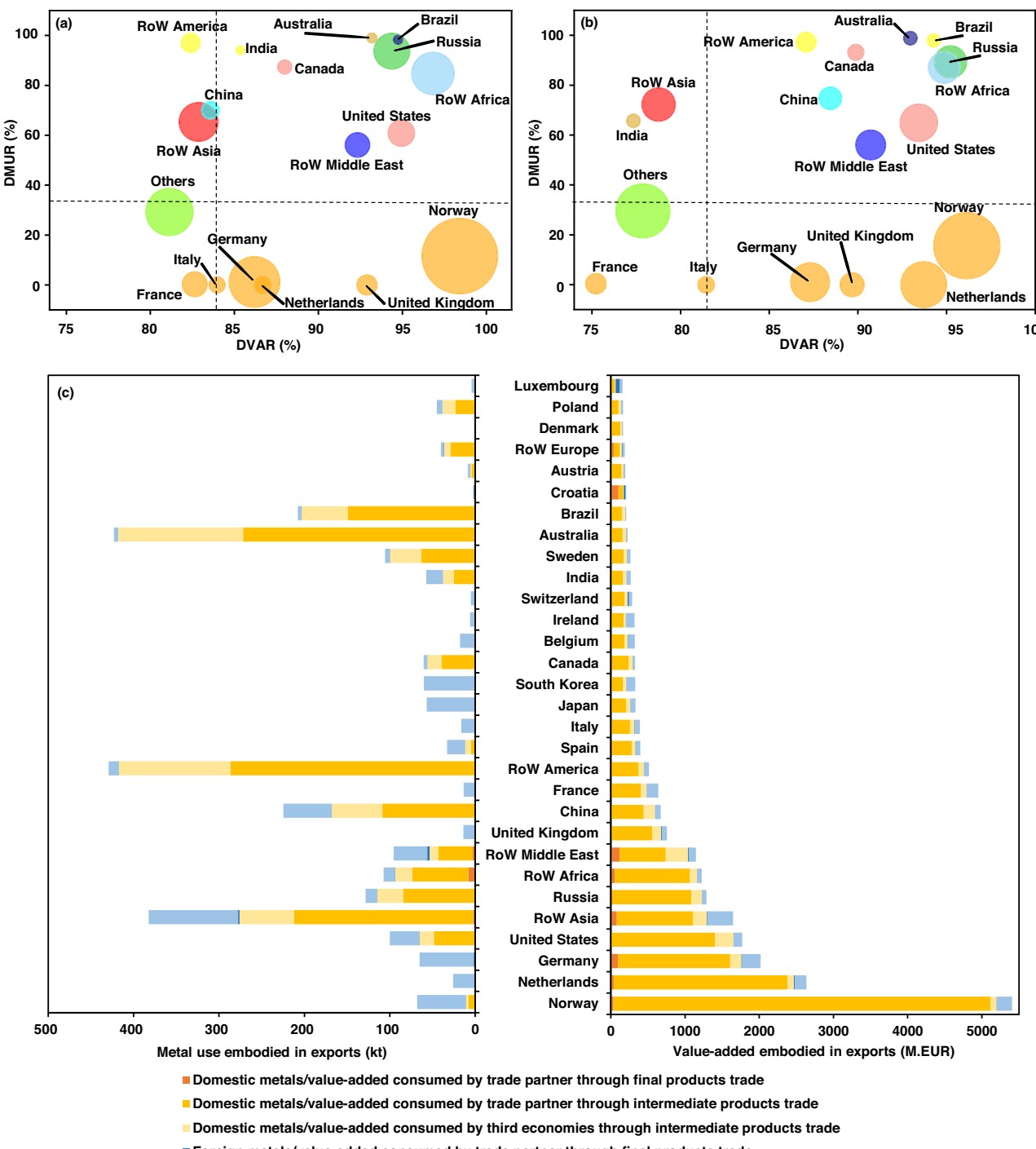

**Fig. 3 | Domestic metal use ratio (DMUR) and domestic value-added ratio (DVAR) of total exports for satisfying the foreign renewable power demand, metal use or value-added embodied in renewable power value chains (RPVCs) for economies.** DMUR and DVAR for different economies in **a** 2005 and **b** 2015. The color and size of the bubble represent distinct economic classifications and the share of an economy's domestic value-added, respectively. The horizontal and vertical lines indicate global average DMUR and DVAR, respectively. Economies with <1% of the global total value-added are aggregated into Others. **c** The metals and value-added (million EUR, M.EUR) embodied in exports through global RPVCs for the largest 30 economies in 2015, with rankings based on their scale of value-added embodied in exports. The details of the remaining economies with <10% of global metal use or value-added embodied in exports are presented in Supplementary Fig. 7.

## Discussion

We comprehensively investigate the MFs and value-added of global and major economies' renewable power sectors to identify the metal product suppliers in renewable power demand. Furthermore, we reveal the imbalances in global RPVCs, in which developing economies support the renewable power demand of developed economies by mining and processing metal products with low economic value. Finally, we present that our results provide valuable information for reasonable and scientific management of metal resources and RPVCs.

The growing MFs inequality along global RPVCs may hinder the just net-zero transition and climate change mitigation actions. Our results indicate that the rapid, clean, and low-carbon power transition

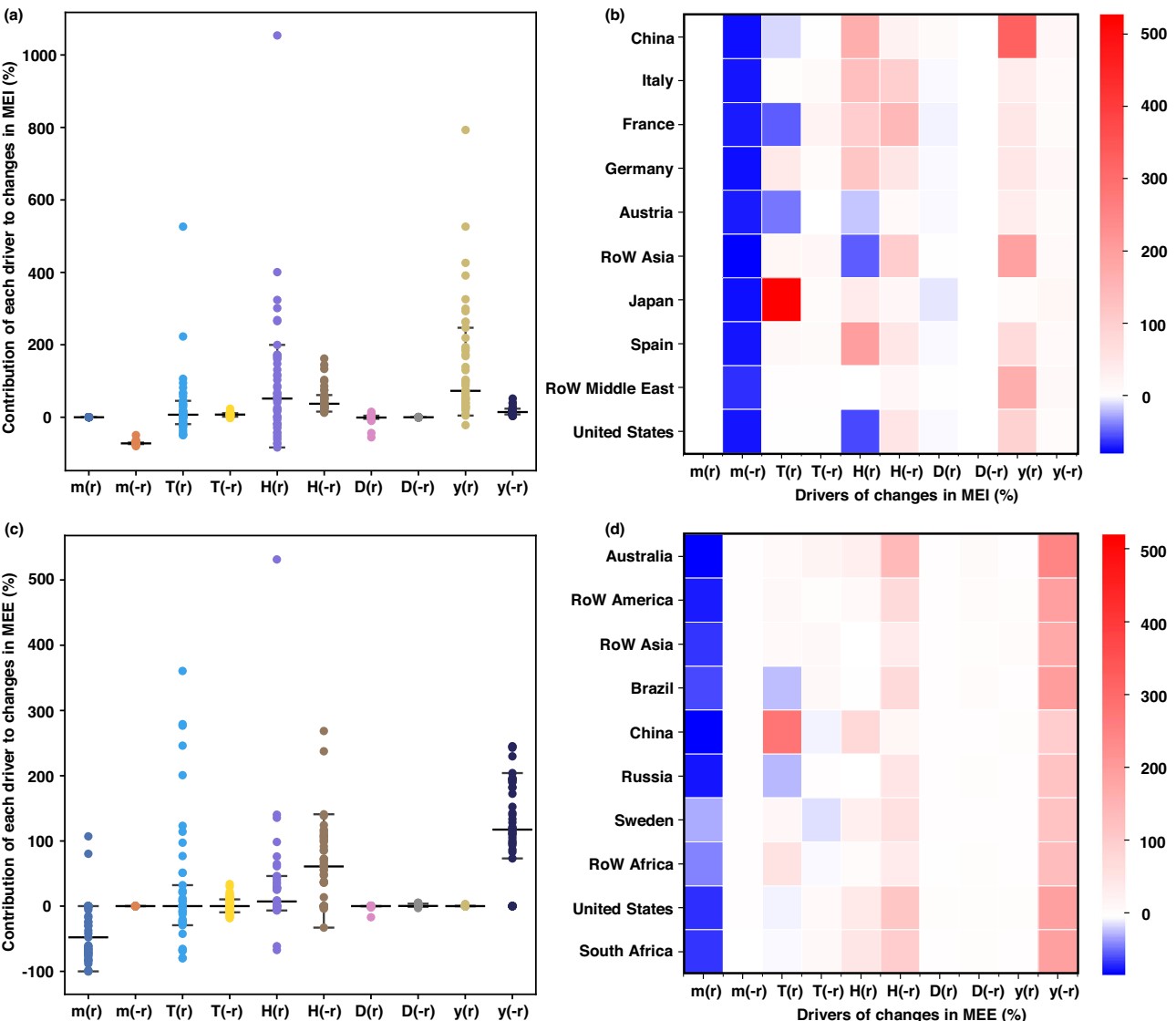

**Fig. 4 | Contribution of each driver to changes in metals embodied in import (MEI) and export (MEE) during 2005–2015. a**, **c** represent changes in MEI and MEE, respectively, with each data point representing 1 of 49 economies. The median and 25th–75th percentiles (bars) are shown. **b**, **d** represent changes in MEI of top 10 importers and MEE of top 10 exporters, sharing 80% of global MEI or MEE, respectively. Additional driver performance of the remaining economies is shown in Supplementary Figs. 9 and 10. **m** represents the direct sectoral metal intensity vector, **T** indicates intermediate product inputs trade structure, **H** indicates production technology, **D** indicates final product trade structure and **y** indicates renewable power demand. Each variable has two parts to distinguish changes at locally (r) or abroad (-r).

in developed economies is built on the ever-growing imports of metal-intensive but low value-added products from developing economies. A recent study revealed that future renewable energy will lead to $PM_{2.5}$ emissions from metal production regionally concentrated in economies such as India and China[52]. Similarly, the displacement of metal mining and production also leads to a shift in greenhouse gas (GHG) emissions to developing economies. For instance, the Democratic Republic of Congo produced ~0.4 Mt of copper for global clean energy technologies, which generated approximately 1 Mt of $CO_2$ emissions, equivalent to 40% of the national total anthropogenic emission in 2020[53–55]. Our results are concurrent with previous literature that these developing economies tend to rely on carbon-intensive metal extraction and mining production technologies under weak environmental regulations and limited climate finance. For example, the $CO_2$ emission intensity of solar PV manufacturing in South Africa (400 kgCO₂/kW) is almost thrice as high as that in Germany (150 kgCO₂/kW)[42].

If no further actions are taken, carbon emissions may be shifted to the primary metal suppliers, thus, impeding the just and timely net-zero transition. In this regard, it is crucial to trace the supply chain environmental performance in RPVCs and incorporate environmental standards into trade policies to promote carbon-efficient production in developing economies[42]. Developed economies could share the responsibility of carbon emission reduction in the minerals field through diverse means, such as low-carbon technology transfer, international climate financial aid expansion, and market-based mechanisms cooperation (such as the Clean Development Mechanism) to stimulate the just net-zero transition[56].

Furthermore, the just net-zero transition and climate goals may also be challenged by the potential metal supply risk[57,58]. Existing evidence indicates that the global metal demand driven by ambitious renewable power expansion cannot not be achieved without a significant production increase, such as a two-fold increase in nickel from 2010 to 2040[10,42]. In contrast, our results indicate that trade conflicts and geopolitical tensions will interrupt future metal supply chains via metal prices[59,60]. As fierce competition aggravates metal scarcity, the net-zero transitions of developing economies such as China, India, Africa, and the Middle East would become uncertain because of metal affordability and availability issues.

Our results highlight an imbalance in the economic benefits and metal product supply of global RPV Cs. The acquisition of clean energy and economic benefits in developed economies usually occurs at the cost of environmental degradation and natural resources depletion in developing economies. For example, in 2015, the United States imported 98% (50 kt) of metal demand for renewable power consumption, leading to a flow of 21 kt of metals from Latin America to the United States. Given that the global power system will transition quickly from fossil fuel-based generation to renewable-resource-based generation, this imbalance may have some key implications for both energy and metal systems. As metal production involves high levels of pollution and environmental emissions, the large-scale development of renewable energy systems will cause serious environmental problems upstream of the industrial chain, potentially leading to overall detrimental effects globally.

To alleviate this imbalance in global RPVCs and its considerable impact on metal supply, strategies aimed at increasing the sustainability of the supply chain on both production and consumption sides should be implemented in parallel. First, technological progresses regarding production can improve the efficiency of metal production, reduce environmental impact, and mitigate metal inequality among economies[61]. Technological innovation for reducing metal intensity and encouraging material substitution in renewable power will play a major role in further metal efficiency improvement. Developed economies, such as the United States and the European Union, have introduced critical material strategies to support Research and Development (R&D) in material efficiency[42]. Considering the mismatch between technology innovation and implementation, technology transfers to accelerate the application of metal-efficient technologies in developing economies is required[62].

In addition to traditional measures, financial tools[63] such as new tax regulations[64] and MFs label certificates[9] (meant to reveal the true cost of embodied metal products) could be enforced for metal mining and producing economies (e.g., South Africa, Congo, China, and Chile). That is, the environmental and health costs of the water, atmosphere, soil pollution, and climate change caused by the extraction, smelting, and transportation of metals that should be considered in the prices of metal products. Additional tax measures would increase the monetary cost of the product and share the cost economically throughout the supply chain. Market behavior can directly encourage producers to reduce production costs[41]; however, the consumption side (the European and American renewable power sectors) can be guided towards metal products with lower environmental costs. Moreover, market selection can assist reduction or even phase out metal products with high production costs or that are non-environmentally friendly.

Notably, there has been some initial activities towards sustainable supply chain management using market tools. The United States and European trade policies emphasize that companies that export photovoltaic modules must issue supply chain traceability certificates[65,66]. In China, a supply chain traceability system for important products is considered an effective measure for supervising the supply chain[67]. In 2019, Changzhou Customs of China applied seven certificates of origin for solar modules exported to Chile, which facilitated $4 million in solar PV sales[68]. Under a bilateral agreement, these goods are expected to elicit more than $200,000 tariff concessions at customs in the importing country[69]. In the future, it is possible that an increase in number of costs and taxes[70] could be incorporated into trade policies based on the consumption of specific metals in the upstream supply chain to help sustainably manage renewable power supply chains.

Changes in the pattern of metal demand may bring new risks to the supply of renewable power, bringing energy security uncertainties. Coal, oil, and natural gas production required for traditional power generation is mainly concentrated in the Middle East and United States[71,72]. Our results further show that the metals required for global renewable power consumption are mainly extracted from Latin America, Africa, and Other Asia. The dependence of renewable power development on raw materials from these regions has reshaped the resource demand pattern in the global power sector.

These changes may bring new risks of metal supply in renewable power development, and the metal flow in our work may help illustrate the supply chain risk source and potential implications. For example, nearly 50% of the metals used in renewable power consumption of European economies originate from Latin America, Africa, and Asia. However, some of these major metal suppliers are faced with uncertain supply policies and geopolitical tensions, which may disrupt the metal supply chain, thereby affecting the stability and resilience of the renewable power market. In 2019, the Indonesian government confirmed that the nickel export ban would be effective by 2020[73]. Price volatility followed and caused the average three-month nickel price to jump by 31% yearly from 2019[74]. Another example is the rise in base metal prices, such as nickel and aluminum, which continued to rise in 2022 due to supply chain disruptions, such as the war between Russia and Ukraine[75]. Consequently, the cost decline of renewable technologies due to technological innovation and economies has mostly reversed. The prices for wind power and solar photovoltaic modules rose by 9 and 16%, respectively. This, in turn, led manufacturers to increase equipment prices, which threatens renewable power expansion schedules[18]. Thus, import-dependent economies need to reduce the dependence on external suppliers and diversify their metal supply to improve the supply self-reliance. In 2010, the United States government formed an interdepartmental working group on strategic mineral supplies for critical metals to improve policies, plans, and procedures to address supply chain risks related to minerals used for renewable power generation, with the goal of diversifying their supply and reducing their heavy reliance on a single economy for metal components used in renewable power generation[76].

In addition, our results indicate that the trade structure can be modified to mitigate metal supply risk and consumption inequality along RPVCs among economies. Import-dependent developed economies can adjust the distribution of traded goods towards metal-efficient sources. For both producers and consumers, trade policies can incorporate resource efficiency standards to select export/import sources, rather than simply transferring metal consumption to downstream economies[77]. For example, China issued guidelines for high-quality trade in 2021 with strict control of the carbon- or energy-intensive products export[78]. Furthermore, it is vital to establish a high-level joint governance framework for standardizing efficiency performance to promote the metal efficiency of the entire supply chain to ensure reliable supply.

Developing economies may face a challenge regarding meeting the metal demand for fast-expanding renewable power infrastructure. According to the International Energy Agency (IEA), the demand for clean power technologies for metal minerals is projected to quadruple by 2040 under sustainable development[42], and 60% of this growth is expected to be driven by developing economies (such as China, India, Brazil)[1,79]. However, our results show that developing economies are more inefficient in the use of metal resources than developed economies. Therefore, improving the efficiency of metal use by reducing loss in primary production and throughout the entire production cycle is crucial in developing economies. Developing economies could save 1041 tons of rare earth metals by 2050 if they increase the efficiency of their metal use in the renewable power sector to its potential level, as determined by the average efficiency level under a net zero emissions scenario[80].

In the long term, effective recycling and reuse is expected to significantly reduce the substantial demand for raw materials and the environmental consequences in developing economies. Currently, there are two approaches to recycling: end-of-life (EOL) recycling and co-

metal recycling. EOL recycling is the most commonly used method. Ninety percent of the base metal materials in the renewable power sector can be recycled through decommissioning, such as by dismantling and disposal of turbine steel, copper, aluminum, and other metals[81,82]. However, the recycling rate is still subject to many factors, such as the depreciation rate of renewable power infrastructure and recycling techniques[10]. Generally, the large-scale installation of wind power started from 2020 and the wind turbines' lifespan usually lasts for approximately 20 years[83], suggesting that there will not be a ramp up of decommissioned metal materials from the recently installed wind turbines until 2040. In addition, recycling during mining and refining is also considered a promising measure to alleviate metal shortages, indicating that waste tailings may become increasingly important in the future[84]. For example, >10 kt of gallium is expected to be recycled from bauxite ores, and more than 15 kt of indium is recycled annually[85,86]. However, these techniques still face significant challenges such as the high cost of tailings, waste collection, and metal loss during the remitting process[87,88]. This suggests that encouraging centralized collection of tailings could be an efficient way to mitigate metal resource waste in the production process, in addition to improving recycling techniques[89].

## Methods

### Metal footprints of renewable power demand

The input-output models used to estimate the MFs are all derived from the classical Leontief equation[90]. This method can identify the input-output relationships between different economies and sectors, as well as the quantity and type of intermediate product inputs required by each economy and sector to produce one unit of output. Using this method, the production process of the final products (e.g., electricity) can be traced.

In a multi-regional input-output (MRIO) model, different economies and sectors are connected through international trade. The technical coefficient matrix $\mathbf{A}$, in which the element $a^{sr}_{nele,ele}$, demonstrates the intermediate inputs of non-electricity (*nele*) sectors in economy $s$ required to produce a unit output of electricity (*ele*) sector in economy $r$. $\mathbf{B} = (\mathbf{I} - \mathbf{A})^{-1}$ denotes the Leontief inverse matrix, which captures both direct and indirect inputs to satisfy per unit of electricity demand. The demand matrix $\mathbf{Y}$, with elements $y^{sr}_{ele}$, indicates the renewable electricity demand in economy $r$ is from economy $s$. To calculate the supply chain metal use embodied in goods and services for renewable power production, we extend MRIO analysis with the metal use as an environmental indicator. $\mathbf{m}$ is a vector of the direct intensity of metal consumption (the volume of domestic metal ores extracted per unit of total output from each sector) for all sectors, for example, $m^s_{nele}$ indicates the direct metal intensity of non-electricity sector in economy $s$. Then the total (including both direct and indirect) metal use embodied in all goods and services for renewable electricity demand via supply chain can be mathematically expressed as follows:

$$\widehat{\mathbf{m}}\mathbf{BY}_{ele} = \begin{bmatrix} m^s_{nele} & 0 & 0 & 0 \\ 0 & m^s_{ele} & 0 & 0 \\ 0 & 0 & m^r_{nele} & 0 \\ 0 & 0 & 0 & m^r_{ele} \end{bmatrix}$$
$$\times \begin{bmatrix} b^{ss}_{nele,nele} & b^{ss}_{nele,ele} & b^{sr}_{nele,nele} & b^{sr}_{nele,ele} \\ b^{ss}_{ele,nele} & b^{ss}_{ele,ele} & b^{sr}_{ele,nele} & b^{sr}_{ele,ele} \\ b^{rs}_{nele,nele} & b^{rs}_{nele,ele} & b^{rr}_{nele,nele} & b^{rr}_{nele,ele} \\ b^{rs}_{ele,nele} & b^{rs}_{ele,ele} & b^{rr}_{ele,nele} & b^{rr}_{ele,ele} \end{bmatrix} \times \begin{bmatrix} 0 & 0 \\ y^{ss}_{ele} & y^{sr}_{ele} \\ 0 & 0 \\ y^{rs}_{ele} & y^{rr}_{ele} \end{bmatrix} \quad (1)$$

where, $\widehat{\mathbf{m}}$ is a matrix with the direct metal intensity for all sectors on the diagonal. We change the demand matrix $\mathbf{Y}$ with zeros for all sectors other than renewable power sectors, namely, production of electricity by hydro, wind, biomass and waste, solar photovoltaic, solar thermal, tide, wave, ocean, and geothermal. The total renewable electricity

demand covers that for both economic production and final demand, such as by households, government, investment, and the coverage applies for the whole analysis.

The metal footprints of renewable power demand of economy $s$ can also be expressed as follows:

$$\mathbf{MF}^s_{ele} = \sum_{r \neq s}^{N} (\widehat{\mathbf{m}^s_c}\mathbf{B}^{ss}\mathbf{y}^{ss} + \widehat{\mathbf{m}^r_c}\mathbf{B}^{rs}\mathbf{y}^{ss} + \widehat{\mathbf{m}^r_c}\mathbf{B}^{rr}\mathbf{y}^{rs} + \widehat{\mathbf{m}^s_c}\mathbf{B}^{sr}\mathbf{y}^{rs}) \quad (2)$$

where, the subscript $c$ represents ten types of metals.

### Metals embodied in trade

Metal embodied in exports (MEE) is expressed as follows according to Xu and Dietzenbacher[91]:

$$\mathbf{MEE}^r = \underbrace{\sum_{s \neq r}^{N} (\widehat{\mathbf{m}^r}\mathbf{B}^{rr}\mathbf{y}^{rs})}_{1} + \underbrace{\sum_{s,k \neq r}^{N} (\widehat{\mathbf{m}^r}\mathbf{B}^{rs}\mathbf{y}^{sk})}_{2} \quad (3)$$

where, the metal embodied in exports can be divided into two parts. The first part represents the metal embodied in economy $r$'s renewable power export that is consumed in another economy. The second part represents the metal embodied in the economy $r$'s intermediate products, which are exported and then used to produce renewable power for consumption for all other economies.

Similarly, metal embodied in import (MEI) is expressed as:

$$\mathbf{MEI}^r = \underbrace{\sum_{s,k \neq r}^{N} (\widehat{\mathbf{m}^k}\mathbf{B}^{ks}\mathbf{y}^{sr})}_{3} + \underbrace{\sum_{s \neq r}^{N} (\widehat{\mathbf{m}^s}\mathbf{B}^{sr}\mathbf{y}^{rr})}_{4} \quad (4)$$

where, the third part represents the global metals embodied in the goods and services imports by economy $s$ to produce renewable power and finally consumed in economy $r$. The fourth part provides the metals in other economies that are embodied in the intermediate products imported by producers in economy $r$ to generate renewable power for consumption.

### Tracing metal use or value-added embodied in global RPVCs

Based on the input-output model, the total bilateral trade of metal use or value-added (export from $s$ to $r$ as example) can be written as:

$$\mathbf{MEEVC}^{sr} = \underbrace{(\widehat{\mathbf{m}^s}\mathbf{B}^{ss})^T \# \mathbf{y}^{sr}}_{1} + \underbrace{(\widehat{\mathbf{m}^s}\mathbf{L}^{ss})^T \# (\mathbf{A}^{sr}\mathbf{B}^{rr}\mathbf{y}^{rr})}_{2}$$
$$+ \underbrace{(\widehat{\mathbf{m}^s}\mathbf{L}^{ss})^T \# \left[ \mathbf{A}^{sr}\mathbf{B}^{rr} \sum_{t \neq s,r}^{G} \mathbf{y}^{rt} + \mathbf{A}^{sr} \sum_{t \neq s,r}^{G} \mathbf{B}^{rt}\mathbf{y}^{tt} + \mathbf{A}^{sr} \sum_{r \neq s,r}^{G} \mathbf{B}^{rt} \sum_{u \neq s,t}^{G} \mathbf{y}^{tu} \right]}_{3}$$
$$+ \underbrace{(\widehat{\mathbf{m}^s}\mathbf{L}^{ss})^T \# \left[ \mathbf{A}^{sr}\mathbf{B}^{rr}\mathbf{y}^{rs} + \mathbf{A}^{sr} \sum_{t \neq s,r}^{G} \mathbf{B}^{rt}\mathbf{y}^{ts} + \mathbf{A}^{sr}\mathbf{B}^{rs}\mathbf{y}^{ss} \right]}_{4}$$
$$+ \underbrace{\left[ (\widehat{\mathbf{m}^s}\mathbf{L}^{ss})^T \# \left( \mathbf{A}^{sr}\mathbf{B}^{rs} \sum_{t \neq s}^{G} \mathbf{y}^{st} \right) + \left( \widehat{\mathbf{m}^s}\mathbf{L}^{ss} \sum_{t \neq s}^{G} \mathbf{A}^{st}\mathbf{B}^{ts} \right)^T \# (\mathbf{A}^{sr}\mathbf{x}^r) \right]}_{5} \quad (5)$$
$$+ \underbrace{(\widehat{\mathbf{m}^r}\mathbf{B}^{rs})^T \# \mathbf{y}^{sr} + \left( \sum_{t \neq s,r}^{G} \widehat{\mathbf{m}^t}\mathbf{B}^{ts} \right)^T \# \mathbf{y}^{sr}}_{6}$$
$$+ \underbrace{\left[ (\widehat{\mathbf{m}^r}\mathbf{B}^{rs})^T \# (\mathbf{A}^{sr}\mathbf{L}^{rr}\mathbf{y}^{rr}) + \left( \sum_{t \neq s,r}^{G} \widehat{\mathbf{m}^t}\mathbf{B}^{ts} \right)^T \# (\mathbf{A}^{sr}\mathbf{L}^{rr}\mathbf{y}^{rr}) \right]}_{7}$$
$$+ \underbrace{\left[ (\widehat{\mathbf{m}^r}\mathbf{B}^{rs})^T \# (\mathbf{A}^{sr}\mathbf{L}^{rr}\mathbf{e}^{r}) + \left( \sum_{t \neq s,r}^{G} \widehat{\mathbf{m}^t}\mathbf{B}^{ts} \right)^T \# (\mathbf{A}^{sr}\mathbf{L}^{rr}\mathbf{e}^{r}) \right]}_{8}$$

Defining "#" as an elementwise matrix multiplication operation, we obtain the total bilateral exports of economy $s$ by summing across the $G$ economies and $N$ sectors, as can be found in Wang and Koopman[92,93] (for a detailed proof, see the Supplementary Section A.1). To clarify the meaning of the eight terms on the right-hand side of the formula, we take metals as an example and provide the following explanations: The first term is the domestic metals of economy $s$ embodied in the final product exports of economy $s$. The second term represents the domestic metals of economy $s$ embodied in intermediate goods exports to $r$, which are used by $r$ to produce final goods that are consumed in $r$. The third term represents the domestic metals embodied in economy $s'$ intermediate exports and used by the direct importing economy $r$ to produce intermediate products that are exported to a third economy $t$ for the production of final consumption goods. The fourth term represents the domestic metals embodied in economy $s'$ exports of intermediate goods used by other economies for their production of final goods that are returned to economy $s$. The fifth term represents a double calculation, that is, the double counting of domestic metals owing to the repeated intermediate goods trade necessary to produce final exports for economy $s$. The sixth term captures the foreign metals used in the final exports of economy $s$. The seventh term indicates the foreign metals used by economy $s$ to produce intermediate goods exports, which are then used by other economies to produce their domestic final goods. The last term represents the foreign metals embodied in intermediate goods exports and used by economy $r$ to produce its intermediate and final goods exports to the world, which are included in the double count of $s'$ exports that originate in foreign economies. $e^{r*}$ is gross exports of economy $r$. Because the double calculation part does not belong to any economy, it is disregarded[19]. To trace the total value-added embodied in global renewable power value chains ($VEEVC^{sr}$), the $m$ vector can be replaced with $v$ vector, which represents the direct value-added coefficients of all sectors.

To obtain the percentage of an economy's domestic metal costs or economic gains in total metal use or value-added embodied in exports for satisfying foreign renewable power demand, two indicators, DMUR and DVAR, were defined and derived as follows:

$$DMUR^s = \sum_{i=1,2,3} MEEVC_i^s \Big/ \sum_{i=1,2,3,6,7} MEEVC_i^s \qquad (6)$$

$$DVAR^s = \sum_{i=1,2,3} VEEVC_i^s \Big/ \sum_{i=1,2,3,6,7} VEEVC_i^s \qquad (7)$$

where, $MEEVC_i^s$ or $VEEVC_i^s (i=1,2,3)$ indicate the domestic parts of total metal use or value-added embodied in exports in Eq. (5), and $MEEVC_i^s$ or $VEEVC_i^s (i=6,7)$ denote the foreign parts.

## Structural decomposition analysis

Structural decomposition analysis (SDA) is widely used to explore the driving force behind changes in resource use or emissions embodied in trade, such as materials resources[94], carbon emissions[95] and mercury emissions[96]. According to Eqs. (12) and (13), MEE (metals embodied in exports) and MEI (metals embodied in imports) depend on the direct sectoral metal intensity vector $m$, input matrix $A$, and demand matrix $Y$[91]. We then decompose the input matrix $A$ into production technology ($H$) and intermediate product input trade structure ($T$). Similarly, the levels of demand ($y$) and final product trade structure ($D$) are used to reflect the demand matrix $Y$. Because of the form of bilateral trade, we divide the five factors into domestic (r) and foreign (-r). The resulting expression for the SDA is as follows:

$$MEE^r = h^r\left(m^{(r)}, m^{(-r)}, T^{(r)}, T^{(-r)}, H^{(r)}, H^{(-r)}, D^{(r)}, D^{(-r)}, y^{(r)}, y^{(-r)}\right) \qquad (8)$$

$$MEI^r = g^r\left(m^{(r)}, m^{(-r)}, T^{(r)}, T^{(-r)}, H^{(r)}, H^{(-r)}, D^{(r)}, D^{(-r)}, y^{(r)}, y^{(-r)}\right) \qquad (9)$$

The first polar is calculated by changing each variable in turn; for example, first changing the first variable, then the second variable, followed by changing the third variable, etc. The second polar is calculated in opposite; we change the last variable first, then the last variable, etc. Further details are provided in the Supplementary Section A.2. The changes in MEE and MEI between year t-1 and t are decomposed by $h_{polar1}^r$ and $h_{polar2}^r$, or $g_{polar1}^r$ and $g_{polar2}^r$, and the geometric average is determined.

$$\Delta MEE_{t-1,t}^r = \frac{MEE_t^r}{MEE_{t-1}^r} = \sqrt{h_{polar1}^r \times h_{polar2}^r} \qquad (10)$$

$$\Delta MEI_{t-1,t}^r = \frac{MEI_t^r}{MEI_{t-1}^r} = \sqrt{g_{polar1}^r \times g_{polar2}^r} \qquad (11)$$

The decomposition of the MEE and MEI changes over a period of years was calculated by multiplying the number of consecutive years. The total change in years 0 to t can be expressed as:

$$\Delta MEE_{0-t}^r = \frac{MEE_t^r}{MEE_0^r} = \frac{MEE_1^r}{MEE_0^r} \times \frac{MEE_2^r}{MEE_1^r} \times \cdots \times \frac{MEE_t^r}{MEE_{t-1}^r}$$

$$= \Delta MEE_{0,t}^r \times \Delta MEE_{1,2}^r \times \cdots \times \Delta MEE_{t-1,t}^r \qquad (12)$$

$$\Delta MEI_{0-t}^r = \frac{MEI_t^r}{MEI_0^r} = \frac{MEI_1^r}{MEI_0^r} \times \frac{MEI_2^r}{MEI_1^r} \times \cdots \times \frac{MEI_t^r}{MEI_{t-1}^r}$$

$$= \Delta MEI_{0,t}^r \times \Delta MEI_{1,2}^r \times \cdots \times \Delta MEI_{t-1,t}^r \qquad (13)$$

## Data sources

There are currently several widely used global multi-regional input-output tables, including EXIOBASE, the World Input-Output Database (WIOD), the Global Trade Analysis Project (GTAP), and Eora, which differ in sectoral and regional resolution[97–100]. We chose the time series EXIOBASE mainly because of its high sectoral resolution (163 sectors, see Supplementary Table 5), including seven renewable power sectors, such as wind power and solar PV[101]. The table covers 44 economies, including 31 European Union member economies and 13 other major ones. The remaining uncovered parts of the world were divided into 5 regions. The currency flows in the multi-regional input-output table are expressed in million EUR. The high-resolution EXIOBASE describes complicated global sectoral linkages between each renewable power sector and all other sectors, allowing us to track direct and indirect metal use or value added along the global RPVCs. Moreover, EXIOBASE facilitates a detailed account of the metal use or value-added of different production stages along RPVCs, thereby revealing the relationships between metal costs and the economic gains of each economy. EXIOBASE is a popular database for revealing the material and other impacts (e.g., emissions) embedded in the global trade of renewable power sectors[35,102,103].

A set of environmental satellite accounts were provided by each sector-region combination and year, which contained metal ores. The selected metals were grouped into four categories, as suggested in the reports of the United Nations Environment Programme (UNEP)

and Word Bank[46,47], including bulk metal ores, precious metal ores, scarce metal ores, and others. Bulk metal ores include bauxite, copper, iron, lead, and zinc ores; precious metal ores include silver and platinum-group metal ores; scarce metal ores include nickel and tin ores; and others include other non-ferrous metal ores. To capture the latest evolution in metals embodied in trade, we use data spanning 2005, 2010, and 2015; all values in 2010 and 2015 were adjusted to the 2005 constant prices (for more data sources, see Supplementary Section A.3). Furthermore, higher levels of disaggregation of metal types and corresponding sectors in EXIOBASE are urgently required, which is crucial for comprehensively understanding how renewable power value chains affect diversified metal consumption worldwide.

## Data availability

All the data used in this study is from open access sources. Specifically, the power capacity and electricity generation data are obtained from the International Renewable Energy Agency (https://www.irena.org)[104]. The Gross National Income (GNI), the national Gross Domestic Product (GDP) and total population data are obtained from the World Bank (https://data.worldbank.org)[105]. The shapefiles used to create maps are from the Environmental Systems Research Institute (https://hub.arcgis.com)[106]. The global MRIO tables are from EXIOBASE (https://www.exiobase.eu/)[101]. The data generated in this study are provided in the Supplementary Information/Source Data file. The metal footprints and value-added data related to renewable power sectors are shown in Supplementary Figs. 1–10 and Tables 3 and 4. Source data underlying all figures in the main manuscript are provided as a Source Data file. Source data are provided with this paper.

## Code availability

The codes of the methods are available at https://doi.org/10.5281/zenodo.7824853[107].

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

## Acknowledgements

This research was financially supported by the National Natural Science Foundation of China (72222014, J.L.; 72348001, J.L.; 72074138, K.F.; 71834004, X.L.), National Social Science Foundation of China (22VMG017, J.L.), and The Key Research & Development Project of Qinghai Province (2022-SF-173, H.Z.).

## Author contributions

R.F., K.P., K.F., and J.L. designed the research. R.F., K.P., P. Z., X.L., B.C., D.C., J.L., and K.F. conducted the analysis. R.F., K.P., P.W., H.Z., B.C., X.L., Y.Z., K.F., and J.L. led the drafting of the manuscript. R.F., K.P., P.W., H.Z., B.C., P. Z., Y.Z., D.C., X.L., K.F., and J.L. contributed to the writing of the article.

## Competing interests

The authors declare no competing interests.
