## [Peer Review file · Nature Communications]

REVIEWER COMMENTS

Reviewer #1 (Remarks to the Author):

General remarks:

The manuscript is good and addresses important issues as it addresses a highly relevant aspect of the world current struggle to shift away from fossil fuels into a new society based on renewables.

However, I do recommend revision as it can and should be improved by making it more concise and clear, as well as strengthening connections to policy and literature.

Secondly, make also sure to follow the guide for authors just to make sure. It mostly looks fine, but there are details to polish on. Many sentences are needlessly long and grammatically complex with many subsegments (one example is starting on line 70 and continues to line 75), potentially subtracting from ease of reading. There are also typos, such as “cooper” instead of “copper”, and various smaller errors to correct here and there.

Finally, I would like to congratulate the authors on an interesting manuscript that is important and look forward to seeing a revised version.

Specific remarks on various subsections:

Introduction:

I like the introduction in general. However, I do think it needs to be revised and make the issue easier for readers to grasp. This can be done by clarifications and by introduction or clarification of relevant terminology.

- Renewable power or renewable energy? Please be clear directly from the start as the opening sentence seems to indicate a focus on electricity from renewables while the paper also covers bioenergy and solar heating via condensing/combined heat/power generation.
- Some chosen words also strike me as a bit unbalance, such as “ultimate solutions” to climate change. Better to call say that it is one of the best options for a more sustainable energy system that would allow society to reduce man-made GHG emissions or meet intended climate goals, such as the Paris Agreement.

- Metals can and should probably be sorted into some better groupings than currently done to avoid confusion and better justify the selected metals for this study. One option could be to define base/bulk metals as geologically common metals, such as iron, copper, aluminium, lead, zinc, etc. that are used for construction of module frames, supporting structures and such. Base metals are typically produced as primary products from mining activities. The second group could be called Scarce metals are geologically less common and rarely form high concentrations due to their geochemical properties. As a result, they are typically recovered as co/by-products to base metals, including metals like titanium, rare earth elements, and much more. Scarce metals typically make up a minor share of the total weight of typical renewable energy technology, but has important functions in electronics, semiconductors, generators, magnets, and such.

- Critical metals are another term that is used in the paper but not clearly defined that can include many different metals depending on specific definitions used to individual studies, where for example EU defines criticality based on high economic importance for the European industry and risk for supply disruptions.

- Precious metals are typically defined as eight specific metals (gold, silver, platinum, palladium, rhodium, ruthenium, iridium, and osmium) that are rare and of high economic value but not all of them are widely used in renewable energy technology. Platina Group Metals (PGMs) are six distinct metals - ruthenium, rhodium, palladium, osmium, iridium, and platinum - with similar chemical properties that typically co-occur together in mineral deposits. So this study define precious metals as silver and PGMs (and excludes gold) as defined in the text for Fig. 1?

- The 10 studied metal groups (Al, Cu, Fe, Pb, Ni, other non-ferrous metals, PGMs, silver, tin, and zinc) strike me as a bit oddly justified. It strikes me as mostly bulk materials (Al, Cu, Pb, Sn, Zn) complemented by a few scarce metals without the proper justification of why they are chosen and other highly relevant materials seemingly excluded (i.e. lithium, cobalt, rare earth elements, etc.). Some of them have been found by previous studies to be critical for low-carbon energy technology (such as silver by Grandell et al (2016)). Some noteworthy references to also consider including are the fresh review by Liang et al (2022) that summarizes previous work done in material requirements for renewables.

- What metals are included in your selection of “other non-ferrous metals”? Titanium, mercury, tungsten, beryllium, bismuth, cerium, cadmium, niobium, indium, gallium, germanium, lithium, selenium, tantalum, tellurium, vanadium, and zirconium?

- Line 93. “Detailed” perhaps sound better than “delicate”?

- It would perhaps be useful to highlight that part of your originality lies in using a more full analysis of the entire value chain than many other studies that just have focused on issues like recoverable resources and expected required volumes arising from future renewables deployment, usually derived from assumed material intensities in selected end-uses in low-carbon energy technologies? Citing and referring to some studies and how your study improves/expands upon them would be useful when presenting research question and briefly presenting the chosen approach?

Results:

I generally like this section and consider the results presented fairly well. Some sentences are a bit clunky, while others are a bit repetitive. The results seem plausible and convincingly and clearly presented.

- Consider standardizing how you present increases. Sometimes it is expressed by growth in percentages, while it is also presented in absolute numbers and increase rates of installed capacity. I suggest consistency for ease of reading.
- Graphs and figures are fine, but I wonder where all 7 renewable power sectors are? What happened to ocean energy and geothermal? Where they too small to merit similar presentations as in Fig 1, so just a simplified presentation as in Fig. S1?
- Abbreviations such as MVAR and DVAR are not properly presented when they first show up (line 0216 and 2017) and may not be seen as self-evident by some readers. Present them the same way as metal footprint (MF) was spelled out and introduced on line 36.
- Figure text in Fig. 3 is a bit repetitive and can be condensed. It can also be good to highlight the most interesting movements of important countries for readers.
- Line 307, 328: change “cooper” to copper.

Discussions;

The discussion and how your results are related to other studies that have explored similar issues or applied similar analysis can be made better. Connect it stronger to existing literature and concept such as resource curse, resource colonialism, political geology, energy security, national self-reliance, geopolitics, or such. Could geopolitical tensions threaten raw material resilience and by extension the possibilities of meeting Paris Agreement and other climate goals? Are these metal footprint inequalities

a growing threat for timely and secure shifts away from fossil fuels? Some references that may be of interest are Breyer et al (2022), Ren et al, (2021), or Troll and Arndt (2022).

References:

Breyer et al, 2022. On the History and Future of 100% Renewable Energy Systems Research. IEEE Access, 10:78176-78218, 2022, doi: 10.1109/ACCESS.2022.3193402.

Grandell et al, 2016. Role of critical metals in the future markets of clean energy technologies. Renewable Energy 95:53-62

Liang et al (2022) Material requirements for low-carbon energy technologies: A quantitative review. Renewable and Sustainable Energy Reviews, Volume 161, June 2022, 112334

Troll and Arndt (2022) European Raw Materials Resilience—Turning a Blind Eye. Earth Sci. Syst. Soc., 19 July 2022. <https://doi.org/10.3389/esss.2022.10058>

Tokimatsu et al, 2017. Energy modeling approach to the global energy-mineral nexus: A first look at metal requirements and the 2 °C target. Applied Energy, 207:494-509

Tokimatsu et al, 2018. Energy modeling approach to the global energy-mineral nexus: Exploring metal requirements and the well-below 2 °C target with 100 percent renewable energy. Applied Energy, 225:1158-1175

Ren et al (2021) Evaluating metal constraints for photovoltaics: perspectives from China's PV development. Applied Energy, 282, Part A: 116148. DOI: <https://doi.org/10.1016/j.apenergy.2020.116148>

Reviewer #2 (Remarks to the Author):

In this paper the authors study the global metals supply chain specifically in case of the renewable power industry. Metal production, import, and export is investigated for developed and developing economies and the economic + environmental impact is evaluated. The authors identify key factors that influence the ever evolving metal footprint in the renewable power supply chains, then conduct a structural decomposition analysis to investigate the impact of the various factors. While the authors offer valuable insights and systemic improvements based on their analysis, some key questions still remain. These questions/comments as well as minor manuscript modifications are highlighted in the attached word document and the authors are requested to address them.

Reviewer #3 (Remarks to the Author):

This paper presents an analysis of metals required for the production of renewable energy technologies. The paper analyses country and regional patterns based on the MRIO database EXIOBASE (49 countries and regions). The topic is very timely and the paper has the potential to be innovative and generating a high impacts.

However, the reviewer still suggests rejecting the paper for publication Nature Communications. The main reasons are that (a) the paper presents unplausible results in several sub-chapters, without the authors reflecting on these results in any way, but rather taking them as granted and (b) the quality of writing is not sufficiently high, in order to easily obtain the main messages that the authors want to convey.

Specific major comments:

- The language the authors use is not suited for a broader audience e.g. in Nature Communications. Already the first paragraph uses a lot of phrases that are difficult to understand or not sufficiently defined: (gearboxes, tower, etc.); to gain added value; (polysilicon, Si-metal, etc.); less developed Asia; etc. This writing style throughout the paper makes it difficult to read. The authors need to be more precise in the terms they use.
- Figure 1: there are very high differences across countries regarding the composition of metals contributing to the total metal footprint of a specific technology. For example, for the case of hydropower, iron ore inputs play the key role in China, but seem to be rather marginal in “ROW America”. Or in the category of wind power, China again uses more than 50% iron ore inputs, whereas copper dominates in Poland and the UK. How can these large differences be explained? A reader would assume that e.g. wind power requires a comparable set of metal ores irrespectively of the country. Is this maybe a data issue related to the uncertainties of sector and product aggregation in EXIOBASE? Have the authors benchmarked their results against existing studies on metal demand of different renewable technologies?
- Figure 2: In this figure, it is not fully clear, what the figure exactly shows. Are the arrows illustrating the extraction of metals to final demand of renewable energy sectors in the consuming country? What does the thick arrow within Europe represent? Which value added is meant when calculating the material intensity? Is it the value added of the extraction sectors? The authors need to be much clearer with regard to defining terms and describing what figures illustrate.
- Figure 3: Regarding this figure, the authors write that the size of the bubble represents the share of each economies’ value added in the total global value added. What does Figure 3 then tell? That Norway holds the largest absolute share in global value added in renewable energy supply chains? This is a very strange message, given that China is leading global production of wind and solar energy. With the same bubble size as France, China seems to be very underrepresented.
- Figure 4: In several parts of the paper, there is no consistency between what is shown in the figure versus written in the text. For example, the panels of Figure 4 include a lot of abbreviations on the x-

axis, but these abbreviations are neither explained in the caption of the figure nor in the text. This makes the results of Figure 4 impossible to read. Which variable refers to final demand, which to metal intensity, i.e. variables that the authors refer to in the text?

- Selection of countries and regions in the figures: Why do the authors not show all countries, but only selections? How did the authors select the countries? Especially in the SI, showing all countries would add transparency without losing “readability”.

- Methods, Data Availability: the authors are much too short in their description of the ability of EXIOBASE to properly capture renewable energy supply chains. Such an evaluation is key, as the whole paper builds on EXIOBASE data. The authors need to explain in detail, e.g.

- o Which sectors of final demand they classified as renewable energy-related?

- o How well is EXIOBASE representing intermediate products in renewable energy supply chains? The authors should provide examples of the supply chain paths that can be modelled with EXIOBASE.

- o Can the sector aggregation of EXIOBASE lead to aggregation errors, when applying it to the case of renewable energies? Could sector aggregation errors explain parts of the unplausible results described above?

- In general, the paper would strongly benefit from native English revision.

Minor points:

Numbers indicate line numbers.

- 70-75: Very long sentence, consider splitting in two.

- 92-93: “... to conduct a more delicate accounting ...” sounds strange. Maybe use “... to apply a more specific accounting ...”

- 106: “... combining a multiregional ...” (otherwise it reads as if there was only one MRIO model available).

- 170: The authors use the term “outsourced metals” without defining this term. Does this term describe metal extraction in foreign countries related to the domestic consumption of renewable technologies?

- 217/218: Is DVAR including the economic gains from metal exports only, or from all exports? It is not fully clear, what is compared here.

- 221, Figure 3: could it be that “Austria” should actually be “Australia”? Because Austria does not have a large domestic metal production section, while Australia has.

- 327-330: What does this result mean? The copper production for renewable energy technologies was responsible for 40% of the 2020 CO2 emissions in Kongo? That is also strange.

- Figure S4: also here, there seems to be a mistake with “Austria”. Austria is listed as the single country with the highest metals embodied in imports and one of the highest countries regarding metals embodied in exports – given the small size of Austria, this is simply impossible.

Reviewer Comments

We would like to thank all the reviewers for taking the time to review our study. Their comments have improved the manuscript substantially. We have revised the whole manuscript carefully, including Abstract, Introduction, Results, Discussion and Methods. Particularly, we did recalculations with the coverage of metal footprints extended from renewable power final demand to total renewable power demand (including power consumption for economic production and final demand), which enables us to trace the metal uses and associated trade network driven by renewable power demand in a more comprehensive and integrated way. Please find the following point-by-point response to all the comments below. All line references are in regard to the revised manuscript. **Text marked in red color** in citations show what part of the text that was revised.

Reviewer #1 (Remarks to the Author):

General remarks:

The manuscript is good and addresses important issues as it addresses a highly relevant aspect of the world current struggle to shift away from fossil fuels into a new society based on renewables.

However, I do recommend revision as it can and should be improved by making it more concise and clear, as well as strengthening connections to policy and literature.

Response: Thanks for your comment. We have revised the whole manuscript carefully to make it more concise and clearer in each section, and enhanced the literature reviews and policies in Introduction and Discussion.

Secondly, make also sure to follow the guide for authors just to make sure. It mostly looks fine, but there are details to polish on. Many sentences are needlessly long and grammatically complex with many subsegments (one example is starting on line 70 and continues to line 75), potentially subtracting from ease of reading.

Response: Thanks for your comment. We have revised the sentence as follows,

(Line 73-78) “With the rapid expansion of renewable power and the increasing complexity of RPVCs, it is increasingly challenging to identify the metal product suppliers of the renewable power materials^{19,20}. Clearly tracing how RPVCs affect metal uses or value-added can provide valuable information for policymakers to formulate trade policies and foster sustainable and responsible

RPVCs.”

In addition, we have checked the whole manuscript to make it more understandable and readable.

There are also typos, such as “cooper” instead of “copper”, and various smaller errors to correct here and there.

Response: Thanks for your comment.

We have corrected “cooper” to “copper” in the revised manuscript and checked and corrected other errors in the whole manuscript carefully and thoroughly.

Finally, I would like to congratulate the authors on an interesting manuscript that is important and look forward to seeing a revised version.

Response: Thank you very much for the positive comment.

Specific remarks on various subsections:

Introduction:

I like the introduction in general. However, I do think it needs to be revised and make the issue easier for readers to grasp. This can be done by clarifications and by introduction or clarification of relevant terminology.

1. Renewable power or renewable energy? Please be clear directly from the start as the opening sentence seems to indicate a focus on electricity from renewables while the paper also covers bioenergy and solar heating via condensing/combined heat/power generation.

Response: Thanks for your comment.

In this article, we focus on renewable power rather than renewable energy. Seven renewable power generation sectors in EXIOBASE were involved, including electricity generation by hydro, wind, biomass and waste, solar photovoltaic, solar thermal, tide, wave, ocean, and geothermal. As for the solar thermal and bioenergy, they represent power generated by converting concentrated sunlight to heat and burning biomass or waste, respectively.

We have modified the relevant expressions in the main text and figures to make the expressions more easily understandable. Additionally, we presented the sector classifications of the reference database, EXIOBASE (163 sectors in total), with the seven renewable power sectors included in the Supplementary **Table S4**).

2. Some chosen words also strike me as a bit unbalance, such as “ultimate solutions” to climate change. Better to call say that it is one of the best options for a more sustainable energy system that would allow society to reduce man-made GHG emissions or meet intended climate goals, such as the Paris Agreement.

Response: Thanks for your comment. We modified the sentences as reviewer suggested,

(Line 62-64) “Renewable power is one of the best options for a more sustainable energy system that would allow society to reduce man-made greenhouse gas emissions or meet intended climate goals, such as the Paris Agreement¹⁻³.”

We also carefully checked through the whole manuscript and rephases any overstated sentences.

3. Metals can and should probably be sorted into some better groupings than currently done to avoid confusion and better justify the selected metals for this study. One option could be to define base/bulk metals as geologically common metals, such as iron, copper, aluminum, lead, zinc, etc. that are used for construction of module frames, supporting structures and such. Base metals are typically produced as primary products from mining activities. The second group could be called Scarce metals are geologically less common and rarely form high concentrations due to their geochemical properties. As a result, they are typically recovered as co/by-products to base metals, including metals like titanium, rare earth elements, and much more. Scarce metals typically make up a minor share of the total weight of typical renewable energy technology, but has important functions in electronics, semiconductors, generators, magnets, and such.

Response: Thanks for your comment and suggestions. In the revised manuscript, we followed reviewer’s suggestion and have grouped all the selected metals into four categories as suggested in the report of United Nations Environment Programme (UNEP) and Word Bank Report^{1,2}, namely bulk metals (i.e., bauxite, copper, iron, lead, and zinc), precious metals (i.e., silver and platinum-group metals), scarce metals (i.e., nickel and tin), and other non-ferrous metals.

We have made the revisions as follows:

(Line 115-121) “Specifically, we focused on widely recognized crucial metals^{31,44,45} for seven renewable power sectors (hydropower, wind power, bioenergy, solar PV, solar thermal, ocean power and geothermal power). The

metals were grouped into four categories, as suggested in the report of the United Nations Environment Programme (UNEP) and Word Bank Report^{46,47}: bulk metal ores (bauxite, copper, iron, lead, and zinc ores), precious metal ores (silver and platinum group metal ores), scarce metal ores (nickel and tin ores), and other non-ferrous metal ores.”

In addition, we clarified the studied metals and the groupings in Data Availability Section, as follows:

(Line 678-685) *“A set of environmental satellite accounts were provided by each sector-region combination and year, which contain metal ores. The selected metals were grouped into four categories, as suggested in the report of the United Nations Environment Programme (UNEP) and Word Bank Report^{46,47}, including bulk metal ores, precious metal ores, scarce metal ores, and others. Bulk metal ores include bauxite, copper, iron, lead, and zinc ores; precious metal ores include silver and platinum-group metal ores; scarce metal ores include nickel and tin ores; and others include other non-ferrous metal ores.”*

References:

1. United Nations Environment Programme, International Resource Panel. Recycling Rates of Metals: A Status Report. <https://wedocs.unep.org/20.500.11822/8702> (2011).
2. Hund, K. et al. Minerals for Climate Action: The Mineral Intensity of the Clean Energy Transition. *World Bank Group* (2020).
4. Critical metals are another term that is used in the paper but not clearly defined that can include many different metals depending on specific definitions used to individual studies, where for example EU defines criticality based on high economic importance for the European industry and risk for supply disruptions.

Response: Thanks for your comments.

Following the reviewer’s suggestion, we have removed the phrase of ‘critical metals’ to avoid confusion. Instead, we classified the selected metals into four different groups (see responses to Comment 3), namely, bulk metal ores, precious metal ores, scarce metal ores and other non-ferrous metal ores.

5. Precious metals are typically defined as eight specific metals (gold, silver, platinum, palladium, rhodium, ruthenium, iridium, and osmium) that are rare and of high

economic value but not all of them are widely used in renewable energy technology. Platina Group Metals (PGMs) are six distinct metals - ruthenium, rhodium, palladium, osmium, iridium, and platinum - with similar chemical properties that typically co-occur together in mineral deposits. So this study defines precious metals as silver and PGMs (and excludes gold) as defined in the text for Fig. 1?

Response: Yes, our study focuses on those (Platina Group Metals and silver) of precious metals, and they are widely considered as critical metal in renewable power technologies¹⁻⁵. Other types of precious metals, e.g., gold is not considered due to the extremely low requirement in renewable power technology. We have clarified those in Introduction Section as follows:

(Line 115-121) “Specifically, we focused on widely recognized crucial metals^{31,44,45} for seven renewable power sectors (hydropower, wind power, bioenergy, solar PV, solar thermal, ocean power and geothermal power). The metals were grouped into four categories, as suggested in the report of the United Nations Environment Programme (UNEP) and Word Bank Report^{46,47}: bulk metal ores (bauxite, copper, iron, lead, and zinc ores), precious metal ores (silver and platinum group metal ores), scarce metal ores (nickel and tin ores), and other non-ferrous metal ores.”

In addition, we clarified the studied metals and the groupings in Data Availability Section, as follows:

(Line 678-685) “A set of environmental satellite accounts were provided by each sector-region combination and year, which contain metal ores. The selected metals were grouped into four categories, as suggested in the report of the United Nations Environment Programme (UNEP) and Word Bank Report^{46,47}, including bulk metal ores, precious metal ores, scarce metal ores, and others. Bulk metal ores include bauxite, copper, iron, lead, and zinc ores; precious metal ores include silver and platinum-group metal ores; scarce metal ores include nickel and tin ores; and others include other non-ferrous metal ores.”

References:

1. Kim, J. et al. Critical and precious materials consumption and requirement in wind energy system in the EU 27. *Appl. Energ.* **139**, 327-334 (2015).
2. Wongnaree, N. et al. Recovery of silver from solar panel waste: an experimental study. In *Materials Science Forum* **1009**, 137-142 (2020).

3. Ren, K., Tang, X., & Höök, M. Evaluating metal constraints for photovoltaics: Perspectives from China's PV development. *Appl. Energy*. **282**, 116148 (2021).
4. Liang, Y. et al. Material requirements for low-carbon energy technologies: A quantitative review. *Renew. Sustain. Energy Rev.* **161**, 112334 (2022).
5. Tokimatsu, K. et al. Energy modeling approach to the global energy-mineral nexus: A first look at metal requirements and the 2°C target. *Appl. Energy* **207**, 494-509 (2017).

6. The 10 studied metal groups (Al, Cu, Fe, Pb, Ni, other non-ferrous metals, PGMs, silver, tin, and zinc) strike me as a bit oddly justified. It strikes me as mostly bulk materials (Al, Cu, Pb, Sn, Zn) complemented by a few scarce metals without the proper justification of why they are chosen and other highly relevant materials seemingly excluded (i.e. lithium, cobalt, rare earth elements, etc.). Some of them have been found by previous studies to be critical for low-carbon energy technology (such as silver by Grandell et al (2016)). Some noteworthy references to also consider including are the fresh review by Liang et al (2022) that summarizes previous work done in material requirements for renewables.

Response: Thanks for the reviewer's comment and suggestions.

First, as we mainly focus on the total metal footprint of renewable technology, those metals with large mass and contribution are captured. While, some critical metals, which are scarce but indispensable for technologies, are not considered in this work. Some metals like rare earth elements¹⁻², chromium and manganese³ are quite relevant to our work but we cannot separate those as they are all included into the type of "other non-ferrous metal" in EXIOBASE. Similarly, some other unnecessary metal components, such as lithium and cobalt mainly used in battery⁴⁻⁷, are also included into the type of "other non-ferrous metal". However, on a (renewable) sector basis, those metals occupied only a small portion of the global metal footprints by 6%, which will not largely affect the results analysis since those primary metals are presented in detail (**Fig. S2**). Still, we acknowledged those limitation, and recommend for a higher level of disaggregation of metal types and corresponding sectors in EXIOBASE in the future.

The revisions in Introduction Section are as follows:

(Line 115-121) "Specifically, we focused on widely recognized crucial metals^{31,44,45} for seven renewable power sectors (hydropower, wind power, bioenergy, solar PV, solar thermal, ocean power and geothermal power). The

metals were grouped into four categories, as suggested in the report of the United Nations Environment Programme (UNEP) and Word Bank Report^{46,47}: bulk metal ores (bauxite, copper, iron, lead, and zinc ores), precious metal ores (silver and platinum group metal ores), scarce metal ores (nickel and tin ores), and other non-ferrous metal ores.”

Moreover, we clarified the studied metals and the groupings in Data Availability Section, as follows:

(Line 678-685) *“A set of environmental satellite accounts were provided by each sector-region combination and year, which contain metal ores. The selected metals were grouped into four categories, as suggested in the report of the United Nations Environment Programme (UNEP) and Word Bank Report^{46,47}, including bulk metal ores, precious metal ores, scarce metal ores, and others. Bulk metal ores include bauxite, copper, iron, lead, and zinc ores; precious metal ores include silver and platinum-group metal ores; scarce metal ores include nickel and tin ores; and others include other non-ferrous metal ores.”*

In addition, we acknowledge the more detailed classifications of metal types as one limitation in our study, as follows:

(Line 687-690) *“Furthermore, higher levels of disaggregation of metal types and corresponding sectors in EXIOBASE are urgently required, which is crucial for comprehensively understanding how renewable power value chains affect diversified metal consumption worldwide.”*

Fig. S2 Global total metal footprints of renewable power demand by metal types in 2015.

References:

1. Fishman, T. & Graedel, T. Impact of the establishment of US offshore wind power on neodymium flows. *Nat. Sustain.* **2**, 332–338. (2019).
2. Liang, Y. et al. Material requirements for low-carbon energy technologies: A quantitative review. *Renew. Sustain. Energy Rev.* **161**, 112334 (2022).
3. Takuma, W. et al. Total material requirement for the global energy transition to 2050: A focus on transport and electricity. *Resour. Conserv. Recy.* **148**, 91-103 (2019).
4. Li, M., & Lu, J. Cobalt in lithium-ion batteries. *Science*, **367**, 979-980 (2020).
5. Grandell, L. et al. Role of critical metals in the future markets of clean energy technologies. *Renew. Energ.* **95**, 53-62. (2016).
6. Tokimatsu, K. et al. Energy modeling approach to the global energy-mineral nexus: A first look at metal requirements and the 2 °C target. *Appl. Energy* **207**, 494-509 (2017).
7. Tokimatsu, K. et al. Energy modeling approach to the global energy-mineral nexus: Exploring metal requirements and the well-below 2 °C target with 100 percent renewable energy. *Appl. Energy* **225**, 1158-1175 (2018).

7. What metals are included in your selection of “other non-ferrous metals”? Titanium, mercury, tungsten, beryllium, bismuth, cerium, cadmium, niobium, indium, gallium, germanium, lithium, selenium, tantalum, tellurium, vanadium, and zirconium?

Response: Thanks for your comment.

Covered by this manuscript, other non-ferrous metals include all the metals mentioned by the reviewer in the revised manuscript since our reference database EXIOBASE covers a full set of metal ores.

8. Line 93. “Detailed” perhaps sound better than “delicate”?

Response: Thanks for your comment. We changed “delicate” to “detailed”, and the sentence is as follows,

(Line 99-100) “Therefore, there is an urgent need to conduct a *detailed* evaluations of both *direct and indirect* metal use or value-added of different production stages along RPVCs...”

9. It would perhaps be useful to highlight that part of your originality lies in using a more full analysis of the entire value chain than many other studies than just have focused on issues like recoverable resources and expected required volumes arising from future renewables deployment, usually derived from assumed material intensities

in selected end-uses in low-carbon energy technologies? Citing and referring to some studies and how your study improves/expands upon them would be useful when presenting research question and briefly presenting the chosen approach?

Response: Thanks for the reviewer’s advice. We have enhanced our contribution as follows:

Most previous studies usually focused on direct metal use estimation for renewable power technologies¹⁻⁴, while the magnitudes of indirect metal use by upstream activities and where it is sourced from were not well captured. In this work, through the use of global multi-regional input-output analysis, we can help fulfil this gap: firstly, we are able to trace and quantify metal footprint, especially the indirect metal uses, associated with seven renewable power sectors in different economies through global supply chains. Thus, we depict a more holistic picture of metal uses and associated trade network driven by renewable power demand. Secondly, with a more detailed decomposition of the metal use or value-added in different renewable power value chain production routes, we can precisely distribute the metal use or value-added to each economy to demonstrate its position. Thus, we can reveal the inequality between metal costs and economic gains among economies in the global renewable power industry.

By doing so, our study offers valuable information on the role of renewable power value chains (*RPVCs*) in embodying and driving the metal costs and economic benefits for each economy. Thus, it can provide useful insights into the reasonable metal use responsibility allocation and government policies to shape responsible and sustainable supply chains.

Accordingly, we cited more references¹⁻¹³, addressed the limitations in current studies and enhanced our contributions in methodologies more clearly. The revisions are as follows:

(Line 81-128) “Currently, some research focuses on estimating the metal demand and constraints of the renewable power sectors. For example, Wang et al. found that the cumulative amount of critical metals required for the production of China’s solar power from 2015 to 2050 will exceed the present national reserve by 1.4—123 folds¹¹ Most previous estimates focused on direct metal use^{10,21,26-32}, while a more comprehensive assessment regarding upstream metal applications related to supply chain activities (such as transportation and service)

is scarce³³⁻³⁵.

...Although previous studies have estimated the region-specific metal demands for renewable power sectors, they lack a detailed and comprehensive picture of the interactions between RPVCs and metal use³⁶. This is because these previous studies generally neglect metal use associated with critical upstream stages or do not identify how production sharing affects metal use or value-added among economies in global RPVCs. Hence, additional efforts are required to trace the true upstream metal costs induced per unit output of renewable power and demonstrate the roles of different economies in global RPVCs^{37,38}. Therefore, there is an urgent need to conduct a detailed evaluations of both direct and indirect metal use or value-added of different production stages along RPVCs, given that crucial information such as the metal costs induced by renewable energy and the position of each economy along RPVCs for reasonably allocating metal use responsibility remains poorly understood^{39,40}

We developed a quantitative framework to gauge metal footprints (MFs, the total metal ores embodied in RPVCs) in global RPVCs by combining a multi-regional input-output model (MRIO) with a value chain decomposition model. The model enables us to track the direct and indirect metal use or value-added associated with all supply chain activities. Specifically, we focused on widely recognized crucial metals^{31,44,45} for seven renewable power sectors (hydropower, wind power, bioenergy, solar PV, solar thermal, ocean power and geothermal power). The metals were grouped into four categories, as suggested in the report of the United Nations Environment Programme (UNEP) and World Bank Report^{46,47}: bulk metal ores (bauxite, copper, iron, lead, and zinc ores), precious metal ores (silver and platinum group metal ores), scarce metal ores (nickel and tin ores), and other non-ferrous metal ores. To the best of our knowledge, we trace the spatial-temporal changes in renewable power sector MFs and value-added in 49 economies during 2005—2015 for the first time. Furthermore, the value chain status of each economy is presented by comparing domestic metal use embodied in exports with the corresponding domestic value-added (see Methods). Consequently, we provide a more holistic view of the growing imbalances in economic benefits and metal costs within RPVCs, highlighting the urgent need to formulate appropriate responsible strategies... ”

References:

1. Wang, P. et al. Incorporating critical material cycles into metal-energy nexus of China's 2050 renewable transition. *Appl. Energy* **253**, 113612 (2019).
2. Watari, T. et al. Total material requirement for the global energy transition to 2050: A focus on transport and electricity. *Resour. Conserv. Recycl.* **148**, 91-103 (2019).
3. Kalt, G. et al. Material requirements of global electricity sector pathways to 2050 and associated greenhouse gas emissions. *J. Clean. Prod.* **358**, 132014 (2022).
4. Li, C. et al. Future material requirements for global sustainable offshore wind energy development. *Renew. Sustain. Energy Rev.* **164**, 112603 (2022).
5. Yang, J. et al. Understanding the material efficiency of the wind power sector in china: a spatial-temporal assessment. *Resour. Conserv. Recycl.* **155**, 104668 (2020).
6. Beylot, A. et al. Mineral raw material requirements and associated climate-change impacts of the French energy transition by 2050. *J. Clean. Prod.* **208**, 1198-1205 (2019).
7. Valero, A. et al. Material bottlenecks in the future development of green technologies. *Renew. Sustain. Energy Rev.* **93**, 178-200 (2018).
8. Ren, K., Tang, X., & Höök, M. Evaluating metal constraints for photovoltaics: Perspectives from China's PV development. *Appl. Energy* **282**, 116148 (2021).
9. Luderer, G. et al. Environmental co-benefits and adverse side-effects of alternative power sector decarbonization strategies. *Nat. Commun.* **10**, 5229 (2019).
10. Gibon, T. et al. A methodology for integrated, multiregional life cycle assessment scenarios under large-scale technological change. *Environ. Sci. Technol.* **49**, 11218-11226 (2015).
11. Hertwich, E. et al. Integrated life-cycle assessment of electricity-supply scenarios confirms global environmental benefit of low-carbon technologies. *P. Natl. Acad. Sci. USA.* **112**, 6277-6282 (2015).
12. Hertwich, E. Increased carbon footprint of materials production driven by rise in investments. *Nat. Geosci.* **14**, 151-155 (2021).
13. Wiedmann, T. et al. The material footprint of nations. *Proc. Natl. Acad. Sci. U. S. A.* **112**, 6271-6276 (2015).

Results:

I generally like this section and consider the results presented fairly well. Some sentences are a bit clunky, while others are a bit repetitive. The results seem plausible and convincingly and clearly presented.

10. Consider standardizing how you present increases. Sometimes it is expressed by growth in percentages, while it is also presented in absolute numbers and increase rates of installed capacity. I suggest consistency for ease of reading.

Response: Thanks for your comment. We unified the expression of increases as absolute numbers and increase rates in the revised manuscript. Detailed revisions are shown below,

(Line 138-141) “Along with the rapid expansion of renewable power infrastructure across the world, the MFs of the global renewable power demand increased by 97% (2425 kt) from 2005 to 2015. Comparatively, the renewable power installed capacity is growing faster, which increased by 125% (1101 GW) over 10 years.”

11. Graphs and figures are fine, but I wonder where all 7 renewable power sectors are? What happened to ocean energy and geothermal? Where they too small to merit similar presentations as in Fig 1, so just a simplified presentation as in Fig. S1?

Response: Thank you and we agree with your comments. We show the metal footprints (MFs) of all 7 renewable power sectors in the modified **Fig. 1** below,

Fig. 1 Metal footprints (MFs) of the renewable power demand by metal types in 2015. Top ten economies in MFs of hydropower (a), wind power (b), bioenergy (c), solar photovoltaic (PV) (d), geothermal power (e), solar thermal (f), ocean power (g). Among all economies, the top ten economies with more than half the global MFs for each

*renewable power demand are included here unless the MFs = 0 (for additional economy MFs, see **Fig. S3**).*

In addition, we supplemented the MFs of the economies not listed in the main text for four renewable power technologies, i.e., hydropower, wind power, bioenergy, solar PV and geothermal power, in the modified **Fig. S3** (shown below),

Fig. S3 Metal footprints (MFs) of the renewable power demand in remaining economies

by metal types in 2015. The bottom 39 economies' MFs in hydropower (a), wind power (b), bioenergy (c), solar photovoltaic (PV) (d) and geothermal power (e). All economies are shown unless the MFs is 0.

12. Abbreviations such as MVAR and DVAR are not properly presented when they first show up (line 0216 and 0217) and may not be seen as self-evident by some readers. Present them the same way as metal footprint (MF) was spelled out and introduced on line 36.

Response: Thanks for your comment.

We have modified the descriptions of MVAR and DVAR and introduced them when they first show up. To be specific:

(Line 248-260) "From a global value chain perspective, exports of goods to other economies may require inputs of both domestic production and imports, thus leads to metal use and value-added generation in both domestic and foreign economies. In this study, we further decompose total metal use or value-added embodied in export (to meet renewable power demand in other economies) into domestic and foreign components to show their distinct positions. Here, we defined two indicators, the domestic shares of total metal exports (MVAR) and total value-added (DVAR), to satisfy foreign renewable power demand (Fig. 3a, 3b and Fig. S6). Higher MVAR or DVAR represents majority of an economy's metal use or value-added triggered by foreign renewable power demand occurs at home. For instance, an economy with a large MVAR but a small DVAR indicates that the economy has a large contribution of metal use to meet foreign renewable power demand but only gain a small economic benefit."

Furthermore, we have added equations to show how to derive MVAR and DVAR in the Method.

(Line 623-630) "To obtain the percentage of an economy's domestic metal costs or economic gains in total metal use or value-added embodied in exports for satisfying foreign renewable power demand, two indicators, MVAR and DVAR, were defined and derived as follows:

$$MVAR^s = \frac{\sum_{i=1,2,3} MEEVC_i^s}{\sum_{i=1,2,3,6,7} MEEVC_i^s} \quad (6)$$

$$DVAR^s = \frac{\sum_{i=1,2,3} VEEVC_i^s}{\sum_{i=1,2,3,6,7} VEEVC_i^s} \quad (7)$$

where, $MEEVC_i^s$ or $VEEVC_i^s$ ($i = 1,2,3$) indicate the domestic parts of total

metal use or value-added embodied in exports in Equation (5), and $MEEVC_i^s$ or $VEEVC_i^s$ ($i = 6,7$) denote the foreign parts.”

13. Figure text in Fig. 3 is a bit repetitive and can be condensed. It can also be good to highlight the most interesting movements of important countries for readers.

Response: Thanks for your comment.

13.1 Figure text in Fig. 3 is a bit repetitive and can be condensed.

We revised the figure text in **Fig. 3** to make it more condensed and also revised the figure to make it more concise, details are as follows,

Fig. 3 Domestic share of total metal export (MVAR) and total value-added (DVAR) for satisfying the foreign renewable power demand, metal use or value-added embodied in renewable power value chains (RPVCs) for economies. MVAR and DVAR for different economies in (a) 2005 and (b) 2015. The color and size of the bubble represent distinct economic classifications and the share of an economy's domestic value-added, respectively. The horizontal and vertical lines indicate global average MVAR and DVAR, respectively. Economies with < 1% of the global total value-added are aggregated into Others. (c) The metals and value-added embodied in exports through global RPVCs for the largest 30 economies in 2015, with rankings based on their scale of value-added embodied in exports. The details of the remaining economies with < 10% of global metal use or value-added embodied in exports are presented in Fig. S7.

13.2 It can also be good to highlight the most interesting movements of important

countries for readers.

Thanks for the valuable suggestion. In the revised version we added description about the interesting movements of important economies, details are as follows.

(Line 301-318) “During 2005—2015, developed economies (such as Norway, Netherlands and Germany) maintained their position in global RPVCs (always with high DVAR) and tend to improve the participation in higher value-added but lower metal-intensive production stages. For instance, with the development of RPVCs, Germany’s DVAR increased by 1.1% (Fig. 3a and 3b), which exhibited a value-added increase of 58 million EUR (0.5% of the total global increase), with the metal embodied in Germany’s exports decreasing by 0.22 kt (Fig. 3c) during the decade concerned. Meanwhile, the developing economies participate in more production activities in global RPVCs, with a slight increase of DVAR and MVAR, which is reflected from the increase in the share of domestic value-added generation or metal use embodied in exports. However, developing economies hold a much larger share in the global total increase of domestic metal exports than that of value-added gains. For example, the metal embodied in Latin America’s exports increased by 154.6 kt (16.5% of the global total increase), only with a light increase in value-added (1.8% of the global total increase). Similarly, with the increase of MVAR and DVAR by 2.7% and 4.9%, respectively, China presents an increase of metals embodied in intermediate goods exports by 94 kt (10% of the global total increase), far larger than the value-added increase (4% of global total increase) between 2005 and 2015.”

14. Line 307, 328: change “cooper” to copper.

Response: Thanks for your comment, we have reviewed and revised the “cooper” to “copper” in the full text.

Discussions;

15. The discussion and how your results are related to other studies that have explored similar issues or applied similar analysis can be made better. Connect it stronger to existing literature and concept such as resource curse, resource colonialism, political geology, energy security, national self-reliance, geopolitics, or such. Could geopolitical tensions threaten raw material resilience and by extension the possibilities of meeting Paris Agreement and other climate goals? Are these metal footprint inequalities a growing threat for timely and secure shifts away from fossil fuels? Some references

that may be of interest are Breyer et al (2022), Ren et al, (2021), or Troll and Arndt (2022).

Response: Thanks for your comment and the recommended references.

15.1 The discussion and how your results are related to other studies that have explored similar issues or applied similar analysis can be made better. Connect it stronger to existing literature.

We linked our results stronger to the existing studies and addressed the different findings derived from our study. Details are as follows,

(Line 378-393) *“The growing MFs inequality along global RPVCs may hinder the just net-zero transition and **climate change mitigation actions**. Our results indicate that the rapid, clean, and low-carbon power transition in developed economies is built on the ever-growing imports of metal-intensive but low value-added products from developing economies. A recent study revealed that future renewable energy will lead to PM_{2.5} emissions from metal production regionally concentrated in economies such as India and China⁵². Similarly, the displacement of metal mining and production also leads to a shift in greenhouse gas (GHG) emissions to developing economies. For instance, the Democratic Republic of Congo produces approximately 0.4 Mt of copper for global clean energy technologies, which generates approximately 1 Mt of CO₂ emissions, equivalent to 40% of the national total anthropogenic emission in 2020⁵³⁻⁵⁵. Our results are concurrent with previous literature that these developing economies tend to rely on carbon-intensive metal extraction and mining production technologies under weak environmental regulations and limited climate finance. For example, the CO₂ emission intensity of solar PV manufacturing in South Africa (400 kgCO₂/kW) is almost thrice as high as that in Germany (150 kgCO₂/kW)⁴⁰.”*

(Line 403-411) *“Furthermore, the just net-zero transition and climate goals may also be challenged by the potential metal supply risk^{57,58}. Existing evidence indicates that the global metal demand driven by ambitious renewable power expansion cannot not be achieved without a significant production increase, such as a two-fold increase in nickel from 2010 to 2040^{10,40}. In contrast, our results indicate that trade conflicts and geopolitical tensions will interrupt future metal supply chains via metal prices^{59,60}. As fierce competition aggravates metal scarcity, the net-zero transitions of developing economies such as China, India, Africa, and the Middle East would become uncertain because of metal*

affordability and availability issues.”

*(Line 504-516) “Developing economies may face a **challenge regarding meeting the metal demand for fast-expanding renewable power infrastructure**. ... However, **our results show that** developing economies are more inefficient in the use of metal resources than developed economies. Therefore, improving the efficiency of metal use by reducing metal loss in primary production and throughout the entire production cycle is crucial in developing economies. Developing economies could save 1,041 tons of rare earth metals by 2050 if they increase the efficiency of their metal use in the renewable power sector to its potential level, as determined by the average efficiency level under a net zero emissions scenario⁸⁰.”*

15.2 Connect it stronger to concept such as resource curse, resource colonialism, political geology, energy security, national self-reliance, geopolitics, or such.

We added the concepts such as political geology, energy security, national self-resilience in the revised manuscript. Details are as follows,

*(Line 463-487) “Changes in the pattern of metal demand may bring new risks to the supply of renewable power, **bringing energy security uncertainties**. Coal, oil, and natural gas production required for traditional power generation is mainly concentrated in the Middle East and United States^{71,72}. Our results further show that the metals required for global renewable power consumption are mainly extracted from Latin America, Africa, and Other Asia. The dependence of renewable power development on raw materials from these regions has reshaped the resource demand pattern in the global power sector.*

*These changes may bring new risks of metal supply in renewable power development, and the metal flow in our work may help illustrate the supply chain risk source and potential implications. For example, nearly 50% of the metals used in renewable power consumption of European economies originate from Latin America, Africa, and Asia. However, some of these major metal suppliers are faced with **uncertain supply policies and geopolitical tensions**, which may disrupt the metal supply chain, thereby affecting the stability and **resilience of the renewable power market**. In 2019, the Indonesian government confirmed that the nickel export ban would be effective by 2020⁷³. Price volatility followed and caused the average three-month nickel price to jump by 31% yearly from 2019⁷⁴.*

Another example is as the rise in base metal price, such as nickel and aluminum, that continued to rise in 2022 due to supply chain disruptions, such as the war between Russia and Ukraine⁷⁵. Consequently, the cost decline of renewable technologies due to technological innovation and economies has mostly reversed. For example, the prices for wind power and solar photovoltaic modules rose by 9% and 16%, respectively. This, in turn, led manufacturers to increase equipment prices, which threatens renewable power expansion schedules¹⁸. Thus, import-dependent economies need to reduce their dependence on external suppliers and diversify their metal supply to improve their metal supply self-reliance....”

15.3 Could geopolitical tensions threaten raw material resilience and by extension the possibilities of meeting Paris Agreement and other climate goals?

We add discussions about how the geopolitical tensions may threaten raw material resilience and by extension the possibilities of meeting Paris Agreement and other climate goals,

(Line 403-411) “Furthermore, the just net-zero transition and climate goals may also be challenged by the potential metal supply risk^{57,58}. Existing evidence indicates that the global metal demand driven by ambitious renewable power expansion cannot not be achieved without a significant production increase, such as a two-fold increase in nickel from 2010 to 2040^{10,40}. In contrast, our results indicate that trade conflicts and geopolitical tensions will interrupt future metal supply chains via metal prices^{59,60}. As fierce competition aggravates metal scarcity, the net-zero transitions of developing economies such as China, India, Africa, and the Middle East would become uncertain because of metal affordability and availability issues.”

15.4 Are these metal footprint inequalities a growing threat for timely and secure shifts away from fossil fuels?

We have strengthened the discussion about how metal footprint inequalities may affect the timely and secure shifts from fossil fuels and climate goals, from two aspects,

- The metal footprint inequalities may hinder the shift from fossil fuels and climate goals due to that the displacement of metal mining and production also

leads to greenhouse gas (GHG) emissions and other environmental issues shift to developing economies¹. If no further actions taken, more carbon emissions may be shifted to the primary metal suppliers, thus, impeding the just and timely net-zero transition². In this regard, we have presented potential measures to help with just net-zero transition.

- The just net-zero transition may be also challenged by the potential metal supply risk, such as sufficient metal supply, and the uncertainty induced by trade conflicts or geopolitical tensions, etc³⁻⁶.

References:

1. Raabe, D. et al. Strategies for improving the sustainability of structural metals. *Nature*, **575**, 64-74 (2019).
2. Tokimatsu, K. et al. Energy modeling approach to the global energy-mineral nexus: Exploring metal requirements and the well-below 2 °C target with 100 percent renewable energy. *Appl. Energy* **225**,1158-1175 (2018).
3. Breyer, C. et al. On the history and future of 100% renewable energy systems research. *IEEE Access*, **10**, 78176-78218 (2022).
4. Lee, J. et al. Reviewing the material and metal security of low-carbon energy transitions. *Renew. Sustain. Energy Rev.* **124**, 109789 (2020).
5. Troll, V. R., & Arndt, N. T. European raw materials resilience—Turning a blind eye. *Earth Sci. Syst. Soc.* **2**, 10058 (2022).
6. Gielen D. Critical minerals for the energy transition. International Renewable Energy Agency: Abu Dhabi, United Arab Emirates, (2021).

Reviewer #2 (Remarks to the Author):

In this paper the authors study the global metals supply chain specifically in case of the renewable power industry. Metal production, import, and export is investigated for developed and developed economies and the economic + environmental impact is evaluated. The authors identify key factors that influence the ever evolving metal footprint in the renewable power supply chains, then conduct a structural decomposition analysis to investigate the impact of the various factors. While the authors offer valuable insights and systemic improvements based on their analysis, some key questions still remain. These questions/comments as well as minor manuscript modifications are highlighted in the attached word document and the authors are requested to address them.

1. Line 33-34. Please add a statement to define the term "RPVC"

Response: Thanks for your comment.

We have added a statement to define the term “RPVC” in both Abstract and Introduction Sections, details are as follows,

(Line 32-37) “Many economies with different endowments and levels of technology participate in various production stages and cultivate value in global renewable power industry production networks, known as global renewable power value chains (RPVCs), complicating the identification of metal supply for the subsequent low-carbon power generation and demand.”

(Line 66-70) “In the current globalized world, economies play different roles in renewable power supply chains ranging from mining, refinery, and component manufacturing, on to the final deployment, and generate revenues (value-added) in each stage of the renewable power value chains (RPVCs)⁷⁻¹⁰.”

2. Line 127. It’s not clear what the inset in Fig. S1 is showing, please explain. Add axes to insets.

Response: Thanks for your comment.

The insets in **Fig. S1** shows the metal footprint (a) and annual installed capacity (b) of geothermal power, solar photovoltaics, solar thermal and ocean power, respectively. As the reviewer suggested, we have added the horizontal and vertical axes titles for the inset in **Fig. S1** and revised the figure text to explain the meaning of the inset clearly. The modified figure is as follows:

Fig. S1 The trends of annual metal footprints (MFs) (a) and installed capacity (b) of renewable power sectors, with geothermal power, solar photovoltaics, solar thermal and ocean power placed in the insets due to the lower value compared with other sectors.

3. Line 134-136. Is the larger size the main factor in MF reduction? What about switch to non-metallic components such as plastics and composites? Please comment.

Response: Thanks for your comment and question.

We fully agree with the reviewer's comments that both size enlargement and materials substitution are major contributors to MFs reduction. We have revised the explanations of MFs reduction as follows,

(Line 144-148) "All results indicate that the installed capacity and electricity generation are growing faster than the MFs of the renewable power demand, mainly because of the material efficiency (t/MW) improvement of renewable power technology in its whole supply chain. There are various options for improving material efficiency, such as design optimization⁴⁷, size enlargement³², and material substitution⁴⁹."

4. Line 142. Can the authors add a figure showing this comment visually please? So the total renewable power contribution % and the MF % for the major contributors.

Response: Thanks for your comment.

Following the reviewer's advice, we have added a panel (c) in **Fig. S1** to visualize the share of total newly added renewable power capacity, electricity generation and MFs for each economy. Accordingly, the revised explanations in main text and figure are as follows,

(Line 151-158) "For example, China is the global leader in renewable power with 67.7 GW a newly added installed capacity (43% of the global total) and 1,381 TWh of renewable electricity generation (24% of the global total), accounting for 61% (Fig. S1 c) of the total MFs of the global renewable power demand. In comparison, the United States, the second-largest global economy, added 17.3 GW of renewable power capacity in 2015 (11% of the global total) and generated 568TWh of renewable electricity (9% of the global total), accounting for only 1% (Fig. S1 c) of the global renewable power sector MFs."

Fig. S1 (c) The shares of MFs, newly added installed capacity, and electricity generation in global total for top 30 economies in 2015, with rankings based on the MFs of all renewable power demand (98% of the global total MFs).

5. Line 145-146. Can the authors add a table with Metal intensity values for countries researched in this paper please?

Response: Thanks for your comment.

We have added a table (**Table S1**) with the metal intensity [total supply chain metal use per unit of renewable electricity generation (kg/MWh)] values for all economies and have shown how the value was derived in Supplementary Section A.3. It shows that the average metal intensity of the renewable power sector in the United States (0.09 kg/MWh) is one-third of that in China (2.22 kg/MWh). The lower metal intensity of the United States is one of the explanations for that the United States has a large installed capacity and electricity generation comparable to China, but its metal footprint is disproportionately smaller than that of China.

Table S1. The full name, abbreviation, GDP per capita (EUR per person) and metal intensity (kg/MWh) (in 2015) of renewable power sector for economies in EXIOBASE.

No.	Abbreviation	Full name	GDP per capita	Metal intensity	Economy categories
1	AUT	Austria	39296	4.41	Europe
2	BEL	Belgium	36462	0.60	Europe
3	BGR	Bulgaria	6276	0.32	Europe
4	CYP	Cyprus	20822	0.42	Middle East
5	CZE	Czech Republic	15860	0.96	Europe
6	DEU	Germany	36547	0.46	Europe
7	DNK	Denmark	47370	0.20	Europe
8	EST	Estonia	15473	1.92	Europe
9	ESP	Spain	22889	0.85	Europe
10	FIN	Finland	38057	0.38	Europe

11	FRA	France	32590	0.97	Europe
12	GRC	Greece	16056	0.04	Europe
13	HRV	Croatia	10480	0.87	Europe
14	HUN	Hungary	11303	1.78	Europe
15	IRL	Ireland	55138	0.38	Europe
16	ITA	Italy	26890	1.13	Europe
17	LTU	Lithuania	12683	1.07	Europe
18	LUX	Luxembourg	90174	18.80	Europe
19	LVA	Latvia	12253	0.55	Europe
20	MLT	Malta	22168	11.98	Europe
21	NLD	Netherlands	40183	2.31	Europe
22	POL	Poland	11189	1.37	Europe
23	PRT	Portugal	17116	1.44	Europe
24	ROM	Romania	7978	0.33	Europe
25	SWE	Sweden	45850	0.32	Europe
26	SVN	Slovenia	18574	0.80	Europe
27	SVK	Slovakia	14509	2.67	Europe
28	GBR	United Kingdom	40062	0.52	Europe
29	USA	United States	50580	0.09	North America
30	JPN	Japan	31097	0.50	Other Asia
31	CHN	China	7176	2.20	China
32	CAN	Canada	38779	0.11	North America
33	KOR	South Korea	25557	0.50	Other Asia
34	BRA	Brazil	7840	0.02	Latin America
35	IND	India	1428	0.20	India
36	MEX	Mexico	8554	0.35	North America
37	RUS	Russia	8284	0.08	Russia
38	AUS	Australia	50484	0.13	Australia
39	CHE	Switzerland	75408	1.19	Europe
40	TUR	Turkey	9790	0.28	Middle East
41	TWN	Taiwan	41767	1.38	China
42	NOR	Norway	66139	0.18	Europe
43	IDN	Indonesia	2964	0.04	Other Asia
44	ZAF	South Africa	5101	0.08	Africa
45	WWA	RoW Asia and Pacific	5795	0.55	Other Asia
46	WWL	RoW America	7677	0.59	Latin America
47	WWE	RoW Europe	6712	0.10	Europe
48	WWF	RoW Africa	1726	1.64	Africa
49	WWM	RoW Middle East	9178	3.00	Middle East

6. Line 204-206. While America is rich in indigenous mineral resources, can the authors comment on the mineral resources available to other large importers such as Europe and China and explain the reason for why these regions are heavy importers?

Response: Thanks for your comment. We have added further comments on the mineral resources available to large importers including Europe and China and explained why they are heavy importers. Concretely,

(Line 204-210) “Europe represented the major importer of metals, with 29%—52% of the total global import volume during the period concerned (Fig. 2a and 2b). In 2015, Europe imported 598 kt, mostly from Latin America (164 kt), Other

Asia (82 kt), and Africa (66 kt). This is mainly attributable to the extreme shortage of indigenous mineral resources, such as copper and lead ores, which cannot meet the large metal demand induced by the huge renewable power industry⁵⁰. As a result, European economies must turn to large metal and renewable power component exporters for imports.”

(Line 213-220) *“Unlike the developed economy, > 63% of the metal consumed by China’s renewable power demand was satisfied by domestic supply (Fig. S5). Although China is rich in several metal endowments (such as zinc), it remains insufficient to meet the fast-growing renewable power industry’s demand, motivated by carbon mitigation ambition⁵¹. In particular, some crucial metals for renewable power technology are extremely deficient, for example, scarce nickel mineral resources with < 5% of global production. As a result, approximately 85% of the nickel footprint of the renewable power demand in China was outsourced from other economies, such as Other Asia.”*

7. Line 290. Sentence wording needs to be fixed.

Response: Thanks for your comment. We have fixed the sentence wording, and details are as follows,

(Line 351-353) *“Meanwhile, majority of the demand ($y^{(r)}$) from developed economies, e.g., European economies, and China induced substantial growth (98%–204%) of MEE for developing economies, such as those in Latin America, Africa, and Other Asia.”*

8. Line 292. Which specific section in the SI?

Response: Thanks for your comment. We have added the specific section in the SI in the main text as follows,

(Line 354-356) *“Comparatively, the changes in production technology ($H^{(r)}$, $H^{(r)}$) and trade structure ($T^{(r)}$) contributed to moderate growth in metal inequality (Figs. S9–10, Supplementary Section B.2).”*

9. Line 297. Please include a table or paragraph in the text or the figure caption to explain the x-axis factors in Fig 4a-d.

Response: Thanks for your comment. We have added the explanations for the x-axis factors in the figure caption in **Fig. 4**, as follows,

(Line 341-344) “***m** represents the direct sectoral metal intensity vector, **T** indicates intermediate product inputs trade structure, **H** indicates production technology, **D** indicates final product trade structure and **y** indicates renewable power demand. Each variable has two parts to distinguish changes at locally (*r*) or abroad (*-r*).*”

10. Line 341. Recent global supply chain issues have affected many economies, if not all, and many metal prices have significantly fluctuated, see especially Lithium-ion battery metal such as Nickel, Lithium, Cobalt, Copper, etc., which have affected renewable energy technologies studied by the authors as well. Can the authors comment on how the presented analysis correlates what we have seen recently?

Response: Thanks for your comment. The comment on recent global supply chain issues particularly the fluctuation of metal prices has been added and highlighted in red color. The comment is as follows,

(Line 403-411) “*Furthermore, the just net-zero transition **and climate goals** may also be challenged by the potential **metal supply risk**^{57,58}. Existing evidence indicates that the global metal demand driven by ambitious renewable power expansion cannot not be achieved without a significant production increase, such as a two-fold increase in nickel from 2010 to 2040^{10,40}. **In contrast, our results indicate that trade conflicts and geopolitical tensions will interrupt future metal supply chains via metal prices**^{59,60}. As fierce competition aggravates metal scarcity, the net-zero transitions of developing economies such as China, India, Africa, and the Middle East would become uncertain because of metal affordability and availability issues.*”

(Line 463-485) “***Changes in the pattern of metal demand may bring new risks to the supply of renewable power, bringing energy security uncertainties.** Coal, oil, and natural gas production required for traditional power generation is mainly concentrated in the Middle East and United States^{71,72}. Our results further show that the metals required for global renewable power consumption are mainly extracted from Latin America, Africa, and Other Asia. The dependence of renewable power development on raw materials from these regions has reshaped the resource demand pattern in the global power sector.*

*...However, some of these major metal suppliers are faced with **uncertain supply policies and geopolitical tensions**, which may disrupt the metal supply chain,*

thereby affecting the stability and resilience of the renewable power market. In 2019, the Indonesian government confirmed that the nickel export ban would be effective by 2020⁷³. Price volatility followed and caused the average three-month nickel price to jump by 31% yearly from 2019⁷⁴. Another example is as the rise in base metal price, such as nickel and aluminum, that continued to rise in 2022 due to supply chain disruptions, such as the war between Russia and Ukraine⁷⁵. Consequently, the cost decline of renewable technologies due to technological innovation and economies has mostly reversed. For example, the prices for wind power and solar photovoltaic modules rose by 9% and 16%, respectively. This, in turn, led manufacturers to increase equipment prices, which threatens renewable power expansion schedules¹⁸.”

Reviewer #3 (Remarks to the Author):

This paper presents an analysis of metals required for the production of renewable energy technologies. The paper analyses economy and regional patterns based on the MRIO database EXIOBASE (49 countries and regions). The topic is very timely and the paper has the potential to be innovative and generating a high impacts.

However, the reviewer still suggests rejecting the paper for publication Nature Communications. The main reasons are that (a) the paper presents unplausible results in several sub-chapters, without the authors reflecting on these results in any way, but rather taking them as granted and (b) the quality of writing is not sufficiently high, in order to easily obtain the main messages that the authors want to convey.

Response: Thanks for your valuable comments and suggestions. With the help of native English speaker, we have revised this manuscript for many runs to avoid grammar errors and further increase its readability. Please see the following point-by-point response to reviewer’s comments.

Specific major comments:

1. The language the authors use is not suited for a broader audience e.g. in Nature Communications. Already the first paragraph uses a lot of phrases that are difficult to understand or not sufficiently defined: (gearboxes, tower, etc.); to gain added value; (polysilicon, Si-metal, etc.); less developed Asia; etc. This writing style throughout the paper makes it difficult to read. The authors need to be more precise in the terms they use.

Response: Thanks for the comment. We have thoroughly revised the language of the

whole manuscript and restate those terms in a more precise manner. The revised paragraph is as follows,

(Line 64-73) “However, the power infrastructure (solar modules, wind turbines, etc.) of renewable power relies on various metals such as iron, copper, aluminum, and other precious metals^{4,5,6}. In the current globalized world, economies play different roles in renewable power supply chains ranging from mining, refinery, and component manufacturing, on to the final deployment, and generate revenues (value-added) in each stage of the renewable power value chains (RPVCs)⁷⁻¹⁰. For instance, solar photovoltaic (PV) value chains use metal ores (copper, aluminum, etc.) from China¹¹ and Africa^{12,13}, and modules (silver, copper, etc.) from Europe¹⁴, the United States¹⁵, and China^{16,17}, which are then assembled in developing economies in Asia, and sold globally¹⁸.”

In addition, we have improved the writing style throughout the paper to make it easy to read, and also ensured that we defined the terms carefully when we used them. Our revised manuscript has been edited by native English language editor, with the Editing Certificate attached.

2. Figure 1: there are very high differences across countries regarding the composition of metals contributing to the total metal footprint of a specific technology. For example, for the case of hydropower, iron ore inputs play the key role in China, but seem to be rather marginal in “ROW America”. Or in the category of wind power, China again uses more than 50% iron ore inputs, whereas copper dominates in Poland and the UK. How can these large differences be explained? A reader would assume that e.g. wind power requires a comparable set of metal ores irrespectively of the economy. Is this maybe a data issue related to the uncertainties of sector and product aggregation in EXIOBASE? Have the authors benchmarked their results against existing studies on metal demand of different renewable technologies?

Response: Thanks for your comment.

For the issues related to Figure 1, indeed, there are large differences across economies regarding the composition of metals in the same power sector. Those differences are mainly because our model captures the total metal footprint of renewable power technology, which includes both direct and indirect (supply chain) metal uses with differentiated upstream sectoral structures in different economies. For example, the indirect metal use of wind power production was mainly from the iron ore-intensive

processes, e.g., electrical machinery and equipment manufacturing sector in China, while by copper ore-intensive processes, e.g., construction sector in Poland.

We agree that the input-output analysis may encounter sectoral aggregation errors, which have been broadly discussed in the existing studies¹. To reduce the uncertainties from sectoral aggregation, our study employs EXIOBASE with 163 individual sectors, including detailed renewable power sectors, which has the highest sector resolution among the current available multi-regional input-output (MRIO) databases². Moreover, the reliability of EXIOBAE in studying the impacts of renewable power sectors from the supply chain perspective have been widely recognized by extensive applications^{3,4}.

The detailed revisions are given as follows,

(Line 166-184) “In general, iron and copper ores were the dominant metals in the production of global renewable power, accounting for 54% and 26% of the total MFs of the global renewable power demand, respectively (Fig. S2), followed by other non-ferrous metals (6% of global MFs). Fig. 1 and Fig. S3 further shows that metal composition of the same renewable power sector is considerably different across economies. For example, iron ores constitute more than half of the MFs of wind power demand in China, whereas copper appears to be the dominant metal in Poland and the United Kingdom. The difference can be attributed to differentiated upstream sectoral structures among the economies (as shown in Fig. S4). For example, indirect metal use contributed to more than 80% of the total metal ore use for per unit of wind power generation in the three economies. In China, iron ore accounts for nearly two-third of the indirect metal use, mainly induced by the electrical machinery, equipment, and fabricated metal products manufacturing sectors, which are iron ore-intensive processes from a supply chain perspective. The United Kingdom and Poland, exhibited a large proportion of copper ores occupancy, mainly induced by the copper ore-intensive processes through the entire supply chain, such as construction or copper mining activities. Likewise, when comparing China with RoW America (Latin American economies, excluding Brazil), hydropower presents distinct patterns, detailed explanations are presented in Fig. S4 and Supplementary Section B.1.”

Fig. S4 The sectoral percentages in economy activities induced by one unit of wind power or hydropower generation for selected economies. (a) the direct sector inputs for wind power or hydropower generation from the first production layer. (b) the indirect sector inputs for wind power or hydropower generation from the second production layer. (c) the total direct and indirect sector inputs for wind power or hydropower generation. For illustrative purpose, only those sectors (see Table S4) closely related to the metal ores consumption are represented, such as mining, manufacturing, construction, and transportation sectors.

Further explanations have been added in the Supplementary Section B.1 as follows,

(Line 181-194) *“We employ the structural path analysis (SPA) to analyze the production network through inter-sector flows, which enables us to investigate the crucial paths for metal ore use driven by renewable power demand along the supply chain^{10,11}. Accordingly, **Fig. S4** depicts the direct and indirect economic activities induced by one unit (i.e., EUR €1) of wind power or hydropower generation from different production layers, which provides information for how the metal ores are used directly or indirectly. For hydropower, the ratio of indirect metal uses to the total metal uses per unit of hydropower generation is more than 60%, in which the iron ore and copper ore accounts for approximately half in China and RoW America (Latin American economies, excluding Brazil), respectively. Specifically, the iron ore consumption is mainly induced by the iron ore-intensive electrical machinery and equipment and fabricated metal products manufacturing sectors in China. Differently, the copper ore consumption was mostly driven by the copper ore-intensive copper mining and construction sectors in RoW America (Latin American economies, excluding Brazil).”*

Our results for metal intensity [total supply chain metal use per unit of electricity generation (kg/MWh)] by different renewable technologies are within the range of results of previous studies, such as Hertwich et al. (2015)⁵, Kleijn et al. (2011)⁶, Barron et al. (2022)⁷, and Guezuraga et al. (2012)⁸. Particularly, the results are comparable to that in Hertwich et al. (2015), which also used EXIOBASE for their analysis. The comparisons between iron, copper, and bauxite intensity in 2010 are presented in **Fig. S12** in the Supplementary Information.

The detailed revisions in the Supplementary Section B.3 are given as follows,

(Line 218-232) *“Although the calculated metal intensity [total supply chain metal use per unit of renewable electricity generation (kg/MWh)] in our study are comparable to that from Hertwich et al. (2015)¹², for some metals such as copper, the intensities are 2-3 times bigger than Hertwich et al. (2015)’s (**Fig. S12**). The differences can be explained by several factors: (a) Differences in the scope of metal. In Hertwich et al. (2015), the metal can be understood as the metal content of the ore extracted and utilized for primary production and of the waste streams utilized for secondary production. In comparison, the extraction of gross ore (run of mine) calculated via the ore grades (or conversion factors) reported for each*

metal in each economy is used in this study. (b) Regional aggregation. Hertwich et al. (2015) constructed a nine-region multi-regional input-output (MRIO) model based on the detailed EXIOBASE MRIO tables. Differently, the detailed EXIOBASE MRIO tables (forty-nine-economy) was employed in this study. The detailed estimation of MFs of each economy can help us gauge the regional heterogeneity and mitigate the impacts of regional aggregation error, resulting in slight differences in the results.”

Fig. S12 The comparisons between our results with that of Hertwich et al. (2015).

References:

1. Su, B. et al. Input-output analysis of CO₂ emissions embodied in trade: The effects of sector aggregation. *Energy Econ.* **32**, 166-175 (2010).
2. Steubing, B. et al. How do carbon footprints from LCA and EEIOA databases compare? A comparison of ecoinvent and EXIOBASE. *J. Ind. Ecol.* **26**, 1406-1422. (2022).
3. Stadler, K. et al. EXIOBASE 3: Developing a time series of detailed environmentally extended multi-regional input-output tables. *J. Ind. Ecol.* **22**, 502–515 (2018).
4. Brockway, P. et al. Estimation of global final-stage energy-return-on-investment for fossil fuels with comparison to renewable energy sources. *Nat Energy* **4**, 612–621 (2019).
5. Hertwich, E. et al. Integrated life-cycle assessment of electricity-supply scenarios confirms global environmental benefit of low-carbon technologies. *P. Natl. Acad. Sci. USA*, **112**, 6277-6282 (2014).
6. Kleijn, R. et al. Metal requirements of low-carbon power generation. *Energy* **36**, 5640-5648 (2011).
7. Barron, K., Hakker, M. & Cullen, J. Material requirements for future low-carbon electricity projections in Africa. *Energy Strateg. Rev.* **44**, 100890 (2022).

8. Guezuraga, B., Zauner, R. & Pölz, W. Life cycle assessment of two different 2 MW class wind turbines. *Renew. Energy* **37**, 37-44, (2012).

3. Figure 2: In this figure, it is not fully clear, what the figure exactly shows. Are the arrows illustrating the extraction of metals to final demand of renewable energy sectors in the consuming country? What does the thick arrow within Europe represent? Which value added is meant when calculating the material intensity? Is it the value added of the extraction sectors? The authors need to be much clearer with regard to defining terms and describing what figures illustrate.

Response: Thanks for your comments, please check the following

3.1 Figure 2: In this figure, it is not fully clear, what the figure exactly shows.

Fig. 2 shows the largest international flows (> 20kt) of metals embodied in renewable power value chains among ten economic groups in 2005 (a) and 2015 (b), with economies shaded according to value of Gross Domestic Product (GDP) per capita.

3.2 Are the arrows illustrating the extraction of metals to final demand of renewable energy sectors in the consuming country?

Yes, the arrows illustrate the extraction of foreign metals driven by the renewable power demand of the consuming economy. For example, as one of the largest flows, the embodied metal exports from Latin America to China reached 352 kt in 2015 through the renewable power value chains, indicating that China's renewable power demand led to 352 kt metal ores production in Latin American economies.

3.3 What does the thick arrow within Europe represent?

The thick arrow within Europe refers to the gross metal embodied in renewable power value chain within Europe. We realized that it may cause confusion if we include it in **Fig. 2**, since the figure is used for presenting the metal flows between 10 groups of economies. Therefore, we delete the arrow within the Europe in the revised **Fig. 2**.

3.4 Which value added is meant when calculating the material intensity? Is it the value added of the extraction sectors?

The metal intensity of each renewable power sector was calculated using the total supply chain metal use divided by the total supply chain value-added of the renewable sector for each economy. The supply chain value-added is generated by all upstream activities, e.g., extraction, transport, manufacturing, construction, and services, for renewable power generation. Therefore, it is the total value added along the entire supply chain of renewable power generation.

While the purpose of the background color is to show that countries at different development levels may play different roles in terms of supplying metals to meet global renewable power development. To make this message clear, we use Gross Domestic Product (GDP) per capita instead of metal intensity as the background color of the maps in the revised manuscript.

To make the figure more easily understood, we added more detailed notes to the figure title as follows,

Fig. 2 Major international flows of metals (>20 kt) embodied in renewable power value chains (RPVCs) among ten groups of economies in 2005 (a) and 2015 (b), with economies shaded according to value of Gross Domestic Product (GDP) per capita.

*The arrows indicate the direction and magnitude of foreign metal ores embodied in renewable power consumed by destination economy. Nearly 80% of global total flows are shown, with the rest flows presented in **Tables S2–3**.*

References:

1. Jiang, M. et al. Provincial and sector-level material footprints in China. *P. Natl. Acad. Sci. USA*. **116**, 26484-26490 (2019).
2. Zhang, W. et al. Unequal exchange of air pollution and economic benefits embodied in China's exports. *Environ. Sci. Technol.* **52**, 3888-3898 (2018).
3. The International Renewable Energy Agency (IRENA), Renewable Energy Statistics 2015, Abu Dhabi. <https://www.irena.org/Data/Downloads/IRENASTAT> (2015).

4. Figure 3: Regarding this figure, the authors write that the size of the bubble represents the share of each economies' value added in the total global value added. What does Figure 3 then tell? That Norway holds the largest absolute share in global value added in renewable energy supply chains? This is a very strange message, given that China is leading global production of wind and solar energy. With the same bubble size as France, China seems to be very underrepresented.

Response: Thanks for your comment, please check the following

4.1 Figure 3: Regarding this figure, the authors write that the size of the bubble represents the share of each economies' value added in the total global value added. What does Figure 3 then tell?

Fig. 3a and 3b both show each economy's position in the renewable power value chain using the domestic metal cost and economic gains to meet foreign renewable power demand. The size of the bubble is based on the ratio of different economies' domestic value-added, that is value-added generated at home, to the global total triggered by renewable power demand of foreign economies. Larger size indicates an economy generates high absolute value-added at home in the global renewable power value chain. In addition, MVAR or DVAR illustrates the percentage of domestic parts in an economy's total metal use or value-added embodied in export to meet foreign economies' renewable power demand. Higher MVAR or DVAR represents majority of an economy's metal-use or value-added triggered by foreign renewable power demand occurs at home. For instance, an economy with a large MVAR but a small DVAR indicates that the economy has a large contribution of metal use to meet foreign renewable power demand but

only gain a small economic benefit. Furthermore, **Fig. 3c** shows metals and value-added embodied in exports through five renewable power value chain routes for major economies, which account for 96% of total global metal use and 94% of global total value-added associated with exports for global renewable power production. For example, the large share of foreign metal consumption in an economy's exports of final goods may indicate that the economy mainly engages in final assembling activities by participating in the low value-added production processes.

We further modified the explanations of **Fig. 3** in both main text and the figure text, details are as follows,

*(Line 247-260) “Each global economy participates in RPVCs at different production stages and generates distinct economic gains with varying metal costs. From a global value chain perspective, exports of goods to other economies may require inputs of both domestic production and imports, thus leads to metal use and value-added generation in both domestic and foreign economies. In this study, we further decompose total metal use or value-added embodied in export (to meet renewable power demand in other economies) into domestic and foreign components to show their distinct positions. Here, we defined two indicators, the domestic shares of total metal exports (MVAR) and total value-added (DVAR), to satisfy foreign renewable power demand (**Fig. 3a, 3b and Fig. S6**). Higher MVAR or DVAR represents majority of an economy's metal use or value-added triggered by foreign renewable power demand occurs at home. For instance, an economy with a large MVAR but a small DVAR indicates that the economy has a large contribution of metal use to meet foreign renewable power demand but only gain a small economic benefit.”*

Fig. 3 Domestic share of total metal export (MVAR) and total value-added (DVAR) for satisfying the foreign renewable power demand, metal use/value-added embodied in renewable power value chains (RPVCs) for economies. MVAR and DVAR for different economies in (a) 2005 and (b) 2015. The color and size of the bubble represent distinct economic classifications and the share of an economy's domestic value-added, respectively. The horizontal and vertical lines indicate global average MVAR and DVAR, respectively. Economies with < 1% of the global total value-added are aggregated into Others. (c) The metals and value-added embodied in exports through global RPVCs for the largest 30 economies in 2015, with rankings based on their scale of value-added embodied in exports. The details of the remaining economies with < 10% of global metal use or value-added embodied in exports are presented in Fig. S7.

Furthermore, to help understand MVAR and DVAR in Fig. 3, we have added equations to show how to derive the two indicators in the Methods.

(Line 623-630) *“To obtain the percentage of an economy’s domestic metal costs or economic gains in total metal use or value-added embodied in exports for satisfying foreign renewable power demand, two indicators, MVAR and DVAR, were defined and derived as follows:*

$$MVAR^s = \frac{\sum_{i=1,2,3} MEEVC_i^s}{\sum_{i=1,2,3,6,7} MEEVC_i^s} \quad (6)$$

$$DVAR^s = \frac{\sum_{i=1,2,3} VEEVC_i^s}{\sum_{i=1,2,3,6,7} VEEVC_i^s} \quad (7)$$

where, $MEEVC_i^s$ or $VEEVC_i^s$ ($i = 1,2,3$) indicate the domestic parts of total metal use or value-added embodied in exports in Equation (5), and $MEEVC_i^s$ or $VEEVC_i^s$ ($i = 6,7$) denote the foreign parts.”

4.2 That Norway holds the largest absolute share in global value added in renewable energy supply chains? This is a very strange message, given that China is leading global production of wind and solar energy. With the same bubble size as France, China seems to be very underrepresented.

The value-added in **Fig. 3** represents an economy's value-added induced by goods and services export to satisfy renewable power demand for another economy, rather than the total value added associated with goods and services for satisfying both domestic and foreign renewable power demand. **Fig. S6** shows an example of the regional value-added distribution of Norway’s gross exports of goods and services to satisfy the United Kingdom’s wind power demand. V1-V7 represent the value added in each production stage, in which the gross exports of Norway = V1+V2+V3+V4+V5+V6, the value-added of Norway = V1+V4+V6, the value-added of China = V3+V5. Therefore, the value-added of Norway (in **Fig. S6**) refers to the value only created in Norway’s production stage in the whole renewable power value chain, which excluded the cost of the upper stage. Accordingly, Norway obtains the largest value-added induced by goods and services export for satisfying foreign renewable power demand (**Fig. 3a and 3b**). As for an economy’s value added created by goods and services to satisfy global (both domestic and foreign) renewable power demand, the global leading renewable power producers (such as China, the United States, and Germany) hold the largest absolute value (**Fig. S8**).

Fig. S6 An example of value-added distribution of Norway’s gross exports of goods and services to satisfy the United Kingdom’s wind power demand.

Fig. S8 Top 30 economies in terms of value added created by goods and services to meet both domestic and foreign renewable power demand. The blue bars represent value added created by goods and services to meet domestic renewable power demand, and the yellow bars represent that to meet foreign renewable power demand.

As mentioned above, **Fig. 3** represents an economy's value-added induced by goods and services export to satisfy renewable power demand for another economy, rather than the total value added associated with goods and services for satisfying both domestic and foreign renewable power demand. The value-added of an economy for satisfying foreign renewable power demand can be determined

by two factors: (1) the export quantity of goods and services; (2) the supply chain value-added created by goods and services in per unit of renewable power generation. On one hand, according to EXIOBASE database, Norway exports considerable goods and services to meet the renewable power demand of foreign economies. For instance, Norway exports a large proportion of total intermediate goods and services used for the renewable power sector, by as high as 98%, for other economy's renewable power production, such as United Kingdom and France in 2015. Comparatively, the value is only 2% in China. On the other hand, the supply chain value-added created by goods and services in per unit of renewable power generation in Norway is more than twice (4.5 EUR/MWh) of the world average (1.8 EUR/MWh), due to that Norway participates in high value-added and metal-efficient production stages in renewable power production, e.g., technology, skill, or service activities (**Fig. S6**). In comparison, China and France present slightly lower value-added created by goods and services in per unit of renewable power generation than Norway with the value of 1.5 EUR/MWh and 1.8 EUR/MWh, respectively. As a result, France and China show similar absolute share in global value-added, sharing 2.6% and 2.1% of the global total, respectively, smaller than that of Norway (23%).

Accordingly, we added the explanations for similar size of the bubble for China and France in the main text as follows,

(Line 292-300) “Interestingly, China, as the world’s top renewable power installer, holds 2.6% of the global total value-added (Fig. 3) induced by goods and services exports to satisfy foreign renewable power demand, similar to France (2.1%), both of which are lower than that of Norway (23%). This is mainly due to the limited scale of goods and services exports in China and their low-end position in production stages, that is, exporting more domestic metals (more than 168 kt in China, Fig. 3c) with less economic gains (> 595 million EUR (M. EUR) in China). In contrast, China occupied the largest value-added created by goods and services to meet both domestic and foreign renewable power demand (Fig. S8).”

5. Figure 4: In several parts of the paper, there is no consistency between what is shown in the figure versus written in the text. For example, the panels of Figure 4 include a lot of abbreviations on the x-axis, but these abbreviations are neither explained in the caption of the figure nor in the text. This makes the results of Figure 4 impossible to

read. Which variable refers to final demand, which to metal intensity, i.e. variables that the authors refer to in the text?

Response: Thanks for your comment.

We have added the descriptions of the abbreviations below **Fig. 4**, and have highlighted it in red color. The explanations are as follows:

*(Line 341-344) “**m** represents the direct sectoral metal intensity vector, **T** indicates intermediate product inputs trade structure, **H** indicates production technology, **D** indicates final product trade structure and **y** indicates renewable power demand. Each variable has two parts to distinguish changes at locally (r) or abroad ($-r$).”*

In addition, to make it easier to follow, we keep the description of the indicators consistent between the main text and the figure in the revised manuscript. Details are as follows,

*(Line 346-358) “We examined the driving forces of the metals embodied in trade to uncover the drivers of growing MFs inequality along global RPVCs (**Fig. 4**). The renewable power demand was the major force of inequality growth. Motivated by the renewable power ambition, the domestic renewable power demand ($y^{(r)}$) boosted MEI growth by 39%–93% in developed economies such as the United States and European economies in 2005–2015. Meanwhile, majority of the demand ($y^{(-r)}$) from developed economies, e.g., European economies, and China induced substantial growth (98%–204%) of MEE for developing economies, such as those in Latin America, Africa, and Other Asia. Comparatively, the changes in production technology ($H^{(r)}$, $H^{(-r)}$) and trade structure ($T^{(r)}$) contributed to moderate growth in metal inequality (**Figs. S9–10, Supplementary Section B.2**). Production technology shifts caused the MEI increase in developed economies (except the United States and Austria) by a wide range of 13%–200%, and MEE growth in developing economies by 14%–117%.”*

6. Selection of countries and regions in the figures: Why do the authors not show all countries, but only selections? How did the authors select the countries? Especially in the SI, showing all countries would add transparency without losing “readability”.

Response: Thanks for your comment, please check the following.

6.1 Selection of countries and regions in the figures: Why do the authors not show all countries, but only selections? How did the authors select the countries?

Fig. 1 shows the top ten economies in terms of metal footprints, accounting for more than half of the global metal footprints for each renewable power demand. **Fig. 2** illustrates the major international flows (>20 kt) of metals embodied in renewable power value chains among top ten groups of economies, sharing nearly 80% of global total flows.

Fig. 3c shows the metals and value-added embodied in exports by different value chain routes for the top 30 economies in terms of value-added embodied in export (> 90 % of global metal use or value-added embodied in export).

Fig. 4b and 4d illustrate the contribution of each factor to the metals embodied in imports (MEI) of top ten importers and the metals embodied in exports (MEE) of top ten exporters, sharing 80% of global MEI or MEE, respectively.

The figures for the rest of economies are shown in Supplementary Information. The explanations are as follows,

In **Fig. 1**, (Line 188-191) *“Among all economies, the top ten economies with more than half the global MFs for each renewable power demand are included here unless the MFs = 0 (for additional economy MFs, see **Fig. S3**).”*

In **Fig. 2**, (Line 233-234) *“Nearly 80% of global total flows are shown, with the rest flows presented in **Tables S2–3**.”*

In **Fig. 3**, (Line 265-273) *“MVAR and DVAR for different economies in (a) 2005 and (b) 2015. The color and size of the bubble represent distinct economic classifications and the share of an economy’s domestic value-added, respectively. The horizontal and vertical lines indicate global average MVAR and DVAR, respectively. Economies with < 1% of the global total value-added are aggregated into Others. (c) The metals and value-added embodied in exports through global RPVCs for the largest 30 economies in 2015, with rankings based on their scale of value-added embodied in exports. The details of the remaining economies with < 10% of global metal use or value-added embodied in exports are presented in **Fig. S7**.”*

In **Fig. 4**, (Line 338-341) *“(b) and (d) represent the contribution of each factor to the MEI of top ten importers and MEE of top ten exporters, sharing 80% of*

*global MEI or MEE, respectively. Additional driver performance of the remaining economies is shown in **Figs. S9–10.***”

6.2 Especially in the SI, showing all countries would add transparency without losing “readability”.

We have presented the metal footprints of hydropower, wind power, bioenergy, solar PV and geothermal power for all the remaining economies in **Figs. S3, S7, S9, S10** (shown below) in the revised version.

Fig. S3 Metal footprints (MFs) of the renewable power demand in remaining economies by metal types in 2015. The bottom 39 economies' MFs in hydropower (a), wind power

(b), bioenergy (c), solar photovoltaic (PV) (d) and geothermal power (e). All economies are shown unless the MFs is 0.

Fig. S7 The metals and value-added embodied in exports through global renewable power value chains for all the economies in 2005 (a) and 2015 (b).

Fig. S9 The contribution of each factor to the metal embodied in export of all economies.

Fig. S10 The contribution of each factor to the metal embodied in import of all economies.

7. Methods, Data Availability: the authors are much too short in their description of the ability of EXIOBASE to properly capture renewable energy supply chains. Such an evaluation is key, as the whole paper builds on EXIOBASE data. The authors need to explain in detail, e.g.

Response: Thanks for your comment. We have enhanced the description of EXIOBASE of its advantages in higher sectoral resolution comparing with other databases. We then clarify its strength in properly quantifying the supply chain metal uses of seven renewable power sectors, and precisely tracing the metal uses in each value chain route in the Methods.

(Line 661-677) “There are currently several widely used global multi-regional input-output tables, including EXIOBASE, the World Input-Output Database

*(WIOD), the Global Trade Analysis Project (GTAP), and Eora, which differ in sectoral and regional resolution⁹⁶⁻¹⁰⁰. We chose the time series EXIOBASE mainly because of its high sectoral resolution (163 sectors, see Supplementary **Table S4**), including seven renewable power sectors, such as wind power and solar PV (<https://www.exiobase.eu/>). The table covers 44 economies, including 31 European Union member economies and 13 other major ones. The remaining uncovered parts of the world were divided into five regions. The currency flows in the multi-regional input–output table are expressed in million euros. The high-resolution EXIOBASE describes complicated global sectoral linkages between each renewable power sector and all other sectors, allowing us to track direct and indirect metal use or value added along the global RPVCs. Moreover, EXIOBASE facilitates a detailed account of the metal use or value-added of different production stages along RPVCs, thereby revealing the relationships between metal costs and the economic gains of each economy. EXIOBASE is a popular database for revealing the material and other impacts (e.g., emissions) embedded in the global trade of renewable power sectors^{35,101,102}. ”*

8. Which sectors of final demand they classified as renewable energy-related?

Response: Thanks for your comment. We clarified the sectors related to demands of renewable power as follows,

*(Line 561-566) “We change the demand matrix **Y** with zeros for all sectors other than renewable power sectors, namely, production of electricity by hydro, wind, biomass and waste, solar photovoltaic, solar thermal, tide, wave, ocean, and geothermal. The total renewable electricity demand covers that for both economic production and final demand, such as by households, government, investment, and the coverage applies for the whole analysis. ”*

In addition, we presented the sector classifications of the reference database, EXIOBASE (163 sectors in total), with the seven renewable power sectors included in the Supplementary **Table S4** to make it more clearly.

9. How well is EXIOBASE representing intermediate products in renewable energy supply chains? The authors should provide examples of the supply chain paths that can be modelled with EXIOBASE.

Response: Thanks for your comment, please check the following

9.1 How well is EXIOBASE representing intermediate products in renewable energy supply chains?

Capturing the total metal footprint with the consideration of intermediate products is always challenging, and the input-output approach is one of the most widely methods for this purpose. There are several widely used global multi-regional input-output databases, such as EXIOBASE, the World Input-Output Database (WIOD), Global Trade Analysis Project (GTAP), and Eora. Among all the currently available tables, EXIOBASE contains seven disaggregated renewable power sectors, while the WIOD, GTAP and Eora databases only have one aggregated power sector¹⁻⁴. It indicates that EXIOBASE has the advantage of illustrating differentiated intermediate inputs to each renewable power sector. Thus, EXIOBASE facilitates us to better capture the metal footprints for each single renewable power demand, mitigating the potential uncertainties induced by the aggregation error of power sector. In addition, EXIOBASE with 163 sectors has the highest sectoral resolution compared with the WIOD, GTAP and Eora databases. Thus, a more detailed interaction between different renewable power sectors and other sectors can be revealed based on EXIOBASE. Hence, we can better trace the metal use or value-added of different production stages for each economy and demonstrate its position in the global renewable power value chains. Finally, EXIOBASE is a popular database and used by many researchers to study the impacts renewable power sectors from the supply chain perspective, such as the environmental benefit⁵, the materials dependence⁶, and employment effects⁷, etc.

Accordingly, we enhanced the description of EXIOBASE in the Data Availability Section,

(Line 661-677) “There are currently several widely used global multi-regional input-output tables, including EXIOBASE, the World Input-Output Database (WIOD), the Global Trade Analysis Project (GTAP), and Eora, which differ in sectoral and regional resolution⁹⁶⁻¹⁰⁰. We chose the time series EXIOBASE mainly because of its high sectoral resolution (163 sectors, see Supplementary Table S4), including seven renewable power sectors, such as wind power and solar PV (<https://www.exiobase.eu/>). The table covers 44 economies, including 31 European Union member economies and 13 other major ones. The remaining uncovered parts of the world were divided into five regions. The currency flows

in the multi-regional input–output table are expressed in million euros. The high-resolution EXIOBASE describes complicated global sectoral linkages between each renewable power sector and all other sectors, allowing us to track direct and indirect metal use or value added along the global RPVCs. Moreover, EXIOBASE facilitates a detailed account of the metal use or value-added of different production stages along RPVCs, thereby revealing the relationships between metal costs and the economic gains of each economy. EXIOBASE is a popular database for revealing the material and other impacts (e.g., emissions) embedded in the global trade of renewable power sectors^{35,101,102}.”

9.2 The authors should provide examples of the supply chain paths that can be modelled with EXIOBASE.

Following the reviewer’s advice, we have provided examples of the supply chains paths that can be modelled with EXIOBASE in the Supplementary Section A.1. The diagram and explanations are as follows,

Fig. S11 Metals uses embodied in an economy’s gross exports by five global renewable power value chain routes.

We combined a multi-regional input-output model (MRIO) with a value chain decomposition model, which enables us to quantify the metal use or value added associated with renewable power sector within each value chain route. Specifically, following Koopman et al. (2014)⁸, and Wang et al. (2013)⁹, the

bilateral trade flows can be decomposed into five parts according to the final consumption destination and product types (intermediate goods with box shaded with grey color and final goods with box shaded with blue color in **Fig. S11**). **Fig. S11** comprehensively shows how an economy's gross exports generate both domestic and foreign metal use or economic gains to satisfy renewable power demand through five global value chain routes, as specified in Equation (8) in the main text.

The five paths can be classified into simple or complex value chains, for which, the traded products cross border only once (paths (1) and (2)), or at least twice (paths (3), (4) and (5)). As an example of the simple value chain, the path (2) in **Fig. S11**, indicates that economy s ' domestic intermediate goods are consumed by direct importer r . An example in the wind power sector is that the Australian's (economy s) iron ore is exported to China (economy r), being manufactured and eventually used in wind farms for electricity generation and consumption in China. For the complex value chains, for example, the path (5) describes that economy s uses the intermediate products from the third economy t to produce the intermediate exports, which is further processed and re-exported to economy r for final consumption. An example for path (5) is that the iron ore is exported from Australia (third economy t) to China (economy s), which is processed into steel in China and then exported to the United States (economy r). The steel is used as inputs for wind tower manufacturing, and finally used for wind power generation and consumption in the United States.

References:

1. Andrew, R. & Peters, G. A multi-region input–output table based on the global trade analysis project database (GTAP-MRIO). *Econ. Syst. Res.* **25**, 99-121 (2013).
2. Dietzenbacher, E. et al. The construction of world input–output tables in the WIOD project. *Econ. Syst. Res.* **25**, 71-98 (2013)
3. Lenzen, M. et al. Building Eora: a global multi-region input–output database at high country and sector resolution. *Econ. Syst. Res.* **25**, 20-49 (2013).
4. Huo, J. et al. Full-scale, near real-time multi-regional input–output table for the global emerging economies (EMERGING). *J. Ind. Ecol.* **26**, 1218-1232 (2022).
5. Hertwich, E. G. et al. Integrated life-cycle assessment of electricity-supply scenarios confirms global environmental benefit of low-carbon technologies. *P. Natl. Acad. Sci. USA*, **112**, 6277-6282 (2015).

6. Kucukvar, M., Onat, N. C., & Haider, M. A. Material dependence of national energy development plans: The case for Turkey and United Kingdom. *J. Clean. Prod.* **200**, 490–500 (2018).
7. Černý, M. et al. Employment Effects of the Renewable Energy Transition in the Electricity Sector. ETUI Research Paper-Working Paper, (2021).
8. Koopman, R., Wang, Z. & Wei, S. Tracing value-added and double counting. *Am. Econ. Rev.* **104**, 459–494 (2014).
9. Wang, Z., Wei, S. & Zhu, K. Quantifying International Production Sharing At the Bilateral and Sector Levels. NBER Working Pap. **19667**, 1–127 (2013).

10. Can the sector aggregation of EXIOBASE lead to aggregation errors, when applying it to the case of renewable energies? Could sector aggregation errors explain parts of the unplausible results described above?

Response: Thanks for the comment. Our results present national average of different renewable power sectors.

We admit that the aggregation inevitably introduces some uncertainties in renewable power value chain, but the applied multi-regional input-output (MRIO) table has been verified to be sufficient to capture the metal use. On the one hand, EXIOBASE database has been widely used to study the impacts of renewable power from the supply chain perspective, such as the environmental benefit¹, the materials dependence², and employment effects³, etc. On the other hand, as broadly discussed in the previous literature⁴⁻⁵, the impacts of aggregation error largely depend on the number and classification of sectors. Compared with The World Input-Output Database (WIOD, 56 sectors), Global Trade Analysis Project (GTAP, 65 sectors), Eora (26 sectors)⁶⁻⁹, the EXIOBASE we applied has the highest sectoral resolution (163 sectors) and the highest renewable power sector resolution (7 renewable power sectors) among the widely used global MRIO databases. Employing the database with highest level of sector disaggregation can minimize aggregate errors in embodied metal accounting¹⁻⁵. For instance, scientific evidence has verified that employing the input-output table with more than 40 sectors is sufficient to capture the overall CO₂ emissions embodied in trade⁵. Hence, using the EXIOBASE in our study, with the highest level of sector disaggregation consisting of 163 sectors, will introduce much smaller uncertainties for the results.

Overall, it is reasonable and sufficient to use EXIOBASE to conduct our analysis.

References:

1. Hertwich, E. G. et al. Integrated life-cycle assessment of electricity-supply scenarios confirms global environmental benefit of low-carbon technologies. *P. Natl. Acad. Sci. USA*, **112**, 6277-6282 (2015).
2. Kucukvar, M., Onat, N. C., & Haider, M. A. Material dependence of national energy development plans: The case for Turkey and United Kingdom. *J. Clean. Prod.* **200**, 490-500 (2018).
3. Černý, M. et al. Employment Effects of the Renewable Energy Transition in the Electricity Sector. ETUI Research Paper-Working Paper, (2021).
4. Lenzen, M. Errors in conventional and Input-Output—based Life—Cycle Inventories. *J. Ind. Ecol.* **4**, 127-148 (2000).
5. Zhang, D. Caron, J. & Winchester, N. Sectoral aggregation error in the accounting of energy and emissions embodied in trade and consumption. *J. Ind. Ecol.* **23**, 402-411. (2019).
6. Andrew, R. & Peters, G. A multi-region input–output table based on the global trade analysis project database (GTAP-MRIO). *Econ. Syst. Res.* **25**, 99-121 (2013).
7. Dietzenbacher, E. et al. The construction of world input–output tables in the WIOD project. *Econ. Syst. Res.* **25**, 71-98 (2013).
8. Lenzen, M. et al. Building Eora: a global multi-region input–output database at high country and sector resolution. *Econ. Syst. Res.* **25**, 20-49 (2013).
9. Huo, J. et al. Full-scale, near real-time multi-regional input–output table for the global emerging economies (EMERGING). *J. Ind. Ecol.* **26**, 1218-1232 (2022).

In general, the paper would strongly benefit from native English revision.

Response: Thanks for highlighting this problem. We have invited native English speakers to help polish the manuscript to improve the language substantially, with the Editing Certificate attached.

Minor points:

Numbers indicate line numbers.

11. 70-75: Very long sentence, consider splitting in two.

Response: Thanks for your comment. The sentence was split into two sentences as follows,

(Line 73-78) “With the rapid expansion of renewable power and the increasing complexity of RPVCs, it is increasingly challenging to identify the metal product

suppliers of the renewable power materials^{19,20}. Clearly tracing how RPVCs affect metal use or value-added can provide valuable information for policymakers to formulate trade policies and foster sustainable and responsible RPVCs.”

12. 92-93: “... to conduct a more delicate accounting ...” sounds strange. Maybe use “... to apply a more specific accounting ...”

Response: Thanks for your comment.

We changed “delicate” to “detailed”, and the revised sentence is as follows,

(Line 99-100) “Therefore, there is an urgent need to conduct a detailed evaluations of both direct and indirect metal use or value-added of different production stages along RPVCs”

13. 106: “... combining a multiregional ...” (otherwise it reads as if there was only one MRIO model available).

Response: Thanks for your comment. The revised sentence is shown below,

(Line 111-113) “We developed a quantitative framework to gauge metal footprints (MFs, the total metal ores embodied in RPVCs) in global RPVCs by combining a multi-regional input-output model (MRIO) with a value chain decomposition model.”

14. 170: The authors use the term “outsourced metals” without defining this term. Does this term describe metal extraction in foreign countries related to the domestic consumption of renewable technologies?

Response: Thank you for this suggestion and the reviewer is correct. We defined the term as follows,

(Line 194-195) “Metal outsourcing indicates that an economy increases metal ore extraction outside its borders for domestic consumption of renewable power.”

15. 217/218: Is DVAR including the economic gains from metal exports only, or from all exports? It is not fully clear, what is compared here.

Response: Thanks for your comment.

DVAR illustrates the percentage of domestic parts in an economy’s total value-added

embodied in export to meet foreign economies' renewable power demand. Domestic economic gains were from all the domestic upstream inputs along the renewable power value chain, consisting of not only the metal extraction sector, but also transportation, manufacturing, and services sectors, etc. Thus, we addressed that we can track direct and indirect value-added associated with all supply chain activities and also revised the description of DVAR to make it clearer,

(Line 113-115) *“The model enables us to track the direct and indirect metal use or value added associated with all supply chain activities.”*

(Line 248-260) *“From a global value chain perspective, exports of goods to other economies may require inputs of both domestic production and imports, thus leads to metal use and value-added generation in both domestic and foreign economies. In this study, we further decompose total metal use or value-added embodied in export (to meet renewable power demand in other economies) into domestic and foreign components to show their distinct positions. Here, we defined two indicators, the domestic shares of total metal exports (MVAR) and total value-added (DVAR), to satisfy foreign renewable power demand (Fig. 3a, 3b and Fig. S6). Higher MVAR or DVAR represents majority of an economy's metal use or value-added triggered by foreign renewable power demand occurs at home. For instance, an economy with a large MVAR but a small DVAR indicates that the economy has a large contribution of metal use to meet foreign renewable power demand but only gain a small economic benefit.”*

16. 221, Figure 3: could it be that “Austria” should actually be “Australia”? Because Austria does not have a large domestic metal production section, while Australia has.

Response: Thanks for highlighting this problem and the reviewer is correct. Australia (18.4% of global total export of metals) is one of the largest exporters of metal ores embodied in goods and services export to satisfy foreign renewable power demand, with rich indigenous mineral resources and significant production. In contrast, Austria has much less export of metals, with less than 0.1% of global total export of metals, as shown in the modified **Fig. 3** (see above Comments 5.1). We checked the whole manuscript thoroughly and carefully to confirm all figures and text correct.

17. 327-330: What does this result mean? The copper production for renewable energy technologies was responsible for 40% of the 2020 CO2 emissions in Kongo? That is also strange.

Response: Thanks for your comment.

This result means that copper production process is CO₂ intensive, and large amount of copper production in Congo is used for renewable power sector, which emits about 40% of national total CO₂ emissions in this economy. It shows that the displacement of metal mining and production also leads to greenhouse gas (GHG) emissions and other environmental issues shift to developing economies. The example indicates that if no further actions taken, more carbon emissions may be shifted to the primary metal suppliers, thus, impeding the just and timely net-zero transition.

18. Figure S4: also here, there seems to be a mistake with “Austria”. Austria is listed as the single country with the highest metals embodied in imports and one of the highest countries regarding metals embodied in exports – given the small size of Austria, this is simply impossible.

Response: Thanks for highlighting this problem and the reviewer is right. As mentioned in Comment 16, Australia is a large exporter of metals, far larger than the export scale of metals from Austria. While Austria is one of the major importers, with 3.3% of global total import of metals, mainly from Russia, Africa and Latin America, etc. The large import is mainly attributable to the extreme shortage of indigenous mineral resources, such as iron, copper and lead ores¹, which cannot meet the metal demand induced by the renewable power generation and demand, accounting for 0.9% of global total in 2015². In comparison, Australia’s total renewable power demand shares only 0.5% of global total², with lower metal import than that of Austria. We have revised **Fig. S7** and checked all the figures and text to confirm they are correct. Details are shown below:

Fig. S7 The metals and value-added embodied in exports through global renewable power value chains for all the economies in 2005 (a) and 2015 (b).

References:

1. BRITISH GEOLOGICAL SURVEY. European Mineral Statistics 2010-14. Nottingham: *British Geological Survey*. (2016).
2. The International Renewable Energy Agency (IRENA), Renewable Energy Statistics 2015, Abu Dhabi. <https://www.irena.org/Data/Downloads/IRENASTAT> (2015).

REVIEWERS' COMMENTS

Reviewer #3 (Remarks to the Author):

Congratulations to the authors for their very comprehensive and solid revision of the manuscript.

The large number of additional explanation, analyses and figures provided make the manuscript now easy to read and interesting for a broad audience.

The reviewer suggest that the paper should be published in its revised form.